# TOPOLOGICAL FLOW MATCHING

**Kacper Wyrwal**[*]
University of Oxford

**İsmail İlkan Ceylan**
TU Wien, AITHYRA, University of Oxford

**Alexander Tong**
AITHYRA

## ABSTRACT

Flow matching is a powerful generative modeling framework, valued for its simplicity and strong empirical performance. However, its standard formulation treats signals on structured spaces—such as fMRI data on brain graphs—as points in Euclidean space, overlooking the rich topological features of their domains. To address this, we introduce *topological flow matching*, a topology-aware generalization of flow matching. We interpret flow matching as a framework for solving a degenerate Schrödinger bridge problem and inject topological information by augmenting the reference process with a Laplacian-derived drift. This principled modification captures the structure of the underlying domain while preserving the desirable properties of flow matching: a stable, simulation-free objective and deterministic sample paths. As a result, our framework serves as a drop-in replacement for standard flow matching. We demonstrate its effectiveness on diverse structured datasets, including brain fMRIs, ocean currents, seismic events, and traffic flows.

## 1 INTRODUCTION

Many of the most valuable datasets in science and engineering do not consist of collections of independent points but are better viewed as signals defined on structured domains— fMRI scans on a brain region graph, ocean current velocities on a mesh, or traffic flows on a road network. The underlying structure of these domains contains crucial information, and yet is often overlooked by standard generative models.

Flow Matching (FM) (Lipman et al., 2023; Liu, 2022; Albergo & Vanden-Eijnden, 2023; Peluchetti, 2023) is a powerful generative modelling framework, achieving state-of-the-art performance across modalities such as images (Esser et al., 2024), video (Polyak et al., 2025), and audio (Vyas et al., 2023), as well as in diverse scientific applications (Tong et al., 2024; Klein et al., 2023). Its appeal lies in a scalable, simulation-free training objective and deterministic sample paths. Despite these advantages, standard FM bears a key limitation: it treats data as points in Euclidean space, neglecting the rich topological and geometric structure of non-Euclidean domains. Prior work in geometric and topological deep learning has shown that respecting such underlying structure can yield substantial performance gains (Bronstein et al., 2021; Papamarkou et al., 2024). Indeed, recent extensions of FM have taken steps in this direction, for example by adapting the framework generating *points* in Riemannian manifolds (Chen & Lipman, 2024) and discrete spaces (Gat et al., 2024). Yet, such ideas have not been utilized in FM for modeling *signals* over such spaces, e.g. fMRI signals on the nodes of a brain graph or current velocities on the edges of a mesh discretizing the ocean.

To fill this gap, we introduce *topological flow matching* (TFM), a principled generalization of FM that exploits the topology of the signal domain. Our key insight is that the relation between FM and the Schrödinger bridge problem (SBP) can be leveraged to inject topological information by augmenting the reference process with a Laplacian-derived drift. Compared to recent work on topological Schrödinger bridges (Yang, 2025), TFM has the distinct advantage of a simulation-free objective and deterministic sample paths. This also makes TFM a seamless substitute for standard FM in applications on structured spaces.

---

[*]Correspondence to `wyrwal.kacper@gmail.com`.
Code available at `https://github.com/KacperWyrwal/topological-flow-matching`.

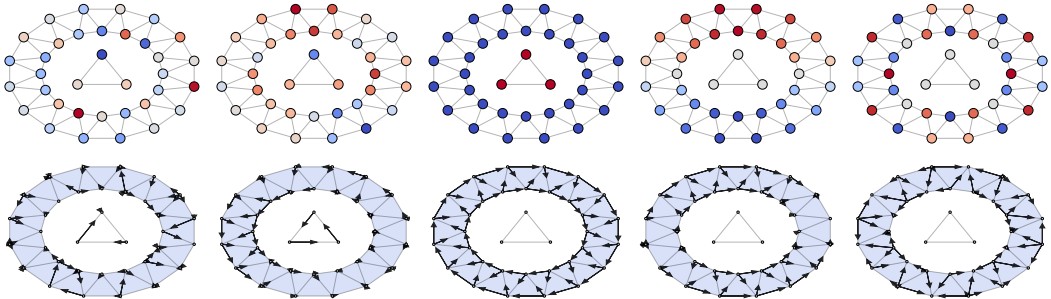

Figure 1: Illustration of the Hodge Laplacian spectrum and its corresponding heat Gaussian process. *Columns from the left*: (1) sample from the normal distribution; (2) sample from the heat Gaussian process; (3) eigenfunction with zero eigenvalue; (4) eigenfunction with low frequency; (5) eigenfunction with high frequency. *Top*: a graph with node signals—low values shown in blue, high in red. *Bottom*: a 2-simplicial complex with edge signals—values proportional to arrow length.

Our contributions are threefold:

- We introduce a principled way of incorporating topological information into FM, utilizing its connection with SBP and augmenting the reference process with a Laplacian-derived drift.

- We derive TFM, a topology-aware generalization of FM for modeling distributions over signals on finite graphs and simplicial complexes. TFM enjoys a stable, simulation-free objective, and deterministic sample paths, making it a drop-in replacement for standard FM.

- We evaluate TFM on diverse datasets on structured domains—including brain fMRI, ocean currents, seismic events, and traffic flows—showing gains over FM and topological Schrödinger bridge matching.

## 2 BACKGROUND

**Notation.** For a stochastic process $X$ with law $\mathbb{P}$ and times $t_0, \ldots, t_m$, we abbreviate $\mathbb{P}((X_{t_0}, \ldots, X_{t_m}) \in \cdot)$ as $\mathbb{P}_{t_0 \ldots t_m}$. Given a random variable $Z$ and a value $z$, we write the conditional distribution $\mathbb{P}(\cdot \mid Z = z)$ as $\mathbb{P}^z$. For a space $\Omega$, we denote the space of probability distributions on $\Omega$ by $\mathcal{P}(\Omega)$. For two distributions $\mu_0, \mu_1 \in \mathcal{P}(\Omega)$ and random variables $X_0 \sim \mu_0, X_1 \sim \mu_1$, the set of couplings of $X_0$ and $X_1$—that is, distributions $\pi \in \mathcal{P}(\Omega \times \Omega)$ with marginals $\pi_0 = \mu_0$ and $\pi_1 = \mu_1$—is denoted $\Pi(\mu_0, \mu_1)$. We denote Brownian motion as $W$. For any vector or matrix, the $i$-th component is indicated by a superscript, e.g. $X_t^i$.

### 2.1 SIGNALS ON STRUCTURED TOPOLOGICAL SPACES

**Graphs.** An undirected graph $K$ consists of nodes $K_0 = \{1, \ldots, n_0\}$ and edges $K_1 = \{[v_0, v_1] : v_0 < v_1 \in K_0\}$ [1]. Denoting $n_k := |K_k|$, the structure of a graph is encoded by its edge-to-node incidence matrix $\boldsymbol{B}_1 \in \mathbb{R}^{n_0 \times n_1}$. The column of $\boldsymbol{B}_1$ corresponding to edge $[v_0, v_1]$ has a $+1$ in the $v_0$-th row, $-1$ in the $v_1$-th row, and $0$ elsewhere. A *node signal* on a graph $K$ is a function $f : K_0 \to \mathbb{R}$, e.g. fMRI data on nodes representing functional regions in a brain graph, identified with its image $f(K_0) \in \mathbb{R}^{n_0}$. An *edge signal* is a function $K_1 \to \mathbb{R}$, e.g. traffic volumes on edges in a road network, identified with $f(K_1) \in \mathbb{R}^{n_1}$.

The *graph Laplacian* $\boldsymbol{L}_0 := \boldsymbol{B}_1 \boldsymbol{B}_1^\top$ is a positive semi-definite matrix acting on node signals. Its eigenvectors with non-zero eigenvalues are wave-like signals with frequencies proportional to their eigenvalues—analogous to sines and cosines for the classical Laplacian on $[0, 1]$. This analogy extends to dynamics: the *graph heat equation* $\dot{f}_t = -\kappa \boldsymbol{L}_0 f_t$, for $\kappa > 0$, describes heat diffusion of an initial node signal $f_0$, while the associated *heat Gaussian process* $\mathcal{N}(0, \exp(-\kappa \boldsymbol{L}_0))$ is a Gaussian distribution over node signals whose covariance reflects the graph structure, reducing to the standard Gaussian if the graph has no edges.

---

[1] An edge $[v_0, v_1]$ is ordered by convention to make the definition of the incidence matrix unambiguous. The same convention extends to simplices in simplicial complexes and their boundary matrices.

**Simplicial complexes.** *Simplicial complexes* model discrete structures more expressively than graphs. A *k-simplicial complex* $K$ consists of nodes $K_0$, edges $K_1$, triangles $K_2 = \{[v_0, v_1, v_2] : v_0 < v_1 < v_2 \in K_0\}$, and so on up to $k$-simplices $K_k = \{[v_0, \ldots, v_k] : v_0 < \cdots < v_k \in K_0\}$. A $k$-simplicial complex can be identified with a polyhedral subspace of $\mathbb{R}^d$, as shown in Figure 1. The structure of a simplicial complex is encoded by the *boundary matrices* $\boldsymbol{B}_k \in \mathbb{R}^{n_{k-1} \times n_k}$, which generalize the edge-to-node incidence matrix. The column of $\boldsymbol{B}_k$ corresponding to a $k$-simplex $[v_0, \ldots, v_k]$ has entries $(-1)^j$ in the rows indexed by its $(k-1)$-faces $[v_0, \ldots, v_{j-1}, v_{j+1}, \ldots, v_k]$, for each $j = 0, \ldots, k$; all other entries are 0. In practice, $K$ may represent an existing real-world structure (e.g., a road network), be specified by domain experts (e.g., a brain-region graph), be constructed from data (e.g., a $k$-nearest-neighbors graph or a Vietoris–Rips complex), or arise from standard geometric constructions in synthetic experiments (e.g., a triangulation of a torus).

A *k-simplex signal* on $K$ is a function $f \colon K_k \to \mathbb{R}$ identified with $f(K_k) \in \mathbb{R}^{n_k}$. The *Hodge Laplacian*

$$\boldsymbol{L}_k \coloneqq \boldsymbol{B}_k^\top \boldsymbol{B}_k + \boldsymbol{B}_{k+1} \boldsymbol{B}_{k+1}^\top$$

is a positive semi-definite matrix, fully determined by the structure of $K$, which acts on $k$-simplex signals, generalizing the graph Laplacian. Its eigenvectors with non-zero eigenvalues correspond to higher-dimensional wave-like signals—e.g., to discrete vector fields for $k = 1$, illustrated in Figure 1. Its associated heat equation $\dot{f}_t = -\kappa \boldsymbol{L}_k f_t$ diffuses signals both through $(k-1)$-simplices via $\boldsymbol{B}_k^\top \boldsymbol{B}_k$ and through $(k+1)$-simplices via $\boldsymbol{B}_{k+1} \boldsymbol{B}_{k+1}^\top$, and also admits a structure-aware heat Gaussian process $\mathcal{N}(0, \exp(-\kappa \boldsymbol{L}_k))$.

**Laplacians and topology.** Eigenvectors of $\boldsymbol{L}_k$ with non-zero eigenvalues are wave-like signals; those with zero eigenvalues reveal *topological* features—intuitively, properties of $K$ preserved under continuous deformations like stretching or twisting, but not discontinuous ones like cutting or gluing[2]. For $k = 0$, these are signals constant on connected components, for $k = 1$, they loop around holes. In general, elements of $\ker \boldsymbol{L}_k$ are signals circulating around "$k$-dimensional holes" called *cohomology classes*. Thus, components of $f \in \mathbb{R}^{n_k}$ in $\ker \boldsymbol{L}_k$ can be viewed as fundamentally aligned with the topological features of $K$.

## 2.2 FLOW MATCHING

Let $\mu_0, \mu_1 \in \mathcal{P}(\mathbb{R}^d)$ be two *boundary distributions*. *Flow matching* (FM) (Lipman et al., 2023; Peluchetti, 2023), also known as rectified flows (Liu, 2022), or stochastic interpolants (Albergo et al., 2025), learns a time-dependent vector field $u \colon [0, 1) \times \mathbb{R}^d \to \mathbb{R}^d$ such that the law $\mathbb{P}$ of the process $X$ driven by the *flow ODE*

$$\dot{X}_t = u_t(X_t), \quad X_0 \sim \mu_0 \tag{1}$$

satisfies $\mathbb{P}_1 = \mu_1$. Samples from $\mu_0$ can be transformed into samples from $\mu_1$ by integrating the ODE, which is used for generation by choosing a simple $\mu_0$ like $\mathcal{N}(0, I_d)$. FM constructs $u$ from *conditional vector fields* $u^z$ driving the conditional process $(X \mid Z = z)$, for a chosen conditioning variable $Z \sim \pi$, as the average:

$$u_t(x) \coloneqq \mathop{\mathbb{E}}_{Z \sim \pi(\cdot \mid X_t = x)} \big[ u_t^Z(x) \big]. \tag{2}$$

Sampling from $\pi(\cdot \mid X_t = x)$ is generally intractable, which makes computation of $u$ via Equation (2) impossible, and also prevents direct minimization of the *FM loss*

$$\mathcal{L}_{\mathrm{FM}}(\theta) \coloneqq \mathop{\mathbb{E}}_{t \sim \mathrm{Unif}[0,1), \ X \sim \mathbb{P}_t} \Big[ \big\| u_t(X) - u_t^\theta(X) \big\|^2 \Big].$$

To overcome this, we need three operations: **(1) evaluation of** $u_t^z(x)$, **(2) sampling** $X_t \sim \mathbb{P}_t^z$, **(3) sampling** $Z \sim \pi$. If we can perform them efficiently, $u$ can be learned by minimizing the *conditional FM loss*

$$\mathcal{L}_{\mathrm{CFM}}(\theta) \coloneqq \mathop{\mathbb{E}}_{t \sim \mathrm{Unif}[0,1), \ Z \sim \pi, \ X \sim \mathbb{P}_t^Z} \Big[ \big\| u_t^Z(X) - u_t^\theta(X) \big\|^2 \Big],$$

---

[2]Formally, $K$ is not a topological space, since it is not equipped with a topology (a collection of open subsets, closed under unions and finite intersections). Nevertheless, we can study its topological features in a meaningful and unambiguous way, since any two polyhedral subspaces that $K$ is identified with are topologically equivalent, or *homeomorphic*.

as guaranteed by the identity $\nabla_\theta \mathcal{L}_{\text{FM}}(\theta) = \nabla_\theta \mathcal{L}_{\text{CFM}}(\theta)$. An especially effective variant of FM, called *conditional flow matching* (CFM) (Lipman et al., 2023), is given by a choice of coupling $\pi \in \Pi(\mu_0, \mu_1)$ and

$$Z = (X_0, X_1), \qquad u_t^{x_0,x_1}(x) = x_1 - x_0.$$

In this case, $(X_t \mid X_0 = x_0, X_1 = x_1)$ follows the straight line $(1-t)x_0 + tx_1$, i.e. $\mathbb{P}_t^{x_0,x_1} = \delta_{(1-t)x_0 + tx_1}$. Two variants of CFM are particularly notable: I-CFM, which uses the independent coupling $\pi = \mu_0 \otimes \mu_1$, and OT-CFM, which uses the *optimal transport* (OT) coupling $\pi^*$ solving the *exact OT problem*

$$\min_{(X_0,X_1)\sim\pi} \mathbb{E}\left[\tfrac{1}{2}\|X_1 - X_0\|^2\right], \quad \text{s.t. } \pi \in \Pi(\mu_0, \mu_1). \tag{3}$$

OT-CFM yields straighter sample paths than I-CFM, which can boost performance (Tong et al., 2024; Pooladian et al., 2023). While powerful, this formulation's connection to the Schrödinger bridge problem, which we introduce next, provides the key to efficiently embedding topological structure in TFM.

## 2.3 THE SCHRÖDINGER BRIDGE PROBLEM

The *Schrödinger bridge problem* (SBP) (Léonard, 2014) with boundary distributions $\mu_0, \mu_1 \in \mathcal{P}(\mathbb{R}^d)$ and a reference law $\mathbb{P}$ over paths $C([0,1]; \mathbb{R}^d)$, with $\mu_0 \otimes \mu_1 \ll \mathbb{P}_{01}$, is the minimization problem

$$\min D_{\text{KL}}(\mathbb{Q}\|\mathbb{P}), \quad \text{s.t. } \mathbb{Q} \in \mathcal{P}(C([0,1]; \mathbb{R}^d)), \ \mathbb{Q} \ll \mathbb{P}, \ \mathbb{Q}_{01} \in \Pi(\mu_0, \mu_1).$$

Intuitively, its solution is the most likely posterior evolution of a system, given a prior belief $\mathbb{P}$, an initial observation $\mu_0$, and a final observation $\mu_1$. It has a unique solution $\mathbb{Q}^*$ given as the mixture

$$\mathbb{Q}^*(E) = \int \mathbb{P}^{x_0,x_1}(E)\mathbb{Q}_{01}^*(\mathrm{d}x_0, \mathrm{d}x_1), \tag{4}$$

implying $\mathbb{Q}^{x_0,x_1} = \mathbb{P}^{x_0,x_1}$. Moreover, if $\mathbb{Q}_{01}^* \ll \mu_0 \otimes \mu_1$, $\mathbb{Q}_{01}^*$ is the solution of the *entropic OT problem*

$$\min_{\mathbb{Q}_{01}} \mathbb{E}[c] + D_{\text{KL}}(\mathbb{Q}_{01}\|\mu_0 \otimes \mu_1), \quad \text{s.t. } \mathbb{Q}_{01} \in \Pi(\mu_0, \mu_1), \tag{5}$$

with the *transport cost* $c = \log \frac{\mathrm{d}\mu_0 \otimes \mu_1}{\mathrm{d}\mathbb{P}_{01}}$. We focus on the SBP with a reference law of the *diffusion SDE*

$$\mathrm{d}X_t = b_t(X_t)\,\mathrm{d}t + \sigma_t(X_t)\,\mathrm{d}W_t, \quad X_0 \sim \mu_0, \tag{6}$$

with a given *drift* $b_t: \mathbb{R}^d \to \mathbb{R}^d$ and *noise* $\sigma_t: \mathbb{R}^d \to \mathbb{R}^{d\times d}$. Heuristically, for small $s > 0$, we have

$$X_{t+s} \approx X_t + b_t(X_t)s + \sigma_t(X_t)\epsilon, \qquad \epsilon \sim \mathcal{N}(0, sI_d).$$

If $\mathbb{P}_t(\cdot \mid X_s = x)$ has the Lebesgue density $p_{s,t}(x, y)$, the conditional process $(X \mid X_0 = x_0, X_1 = x_1)$ follows the *diffusion bridge SDE*

$$\mathrm{d}X_t = [b_t(X_t) + u_t^{x_0,x_1}(X_t)]\,\mathrm{d}t + \sigma_t(X_t)\,\mathrm{d}W_t, \quad u_t^{x_0,x_1}(x) \coloneqq \sigma_t(x)\sigma_t^\top(x)\nabla \log p_{t,1}(x, x_1). \tag{7}$$

Proposition 1 shows that the SBP solution follows an SDE with a drift expressed as a mixture of *conditional controls* $u^{x_0,x_1}$.

**Proposition 1.** *Let $\mathbb{P}$ be the law of a diffusion process $X$ and define the* marginal control $u$ *by*

$$u_t(x) \coloneqq \mathbb{E}_{(X_0,X_1)\sim\mathbb{Q}_{01}^*(\cdot|X_t=x)}\left[u_t^{X_0,X_1}(x)\right]. \tag{8}$$

*The solution $\mathbb{Q}^*$ to the Schrödinger bridge problem with reference law $\mathbb{P}$ is the law of the process*

$$\mathrm{d}X_t = [b_t(X_t) + u_t(X_t)]\,\mathrm{d}t + \sigma_t(X_t)\,\mathrm{d}W_t, \quad X_0 \sim \mu_0.$$

# 3 INCORPORATING TOPOLOGY INTO FLOW MATCHING

In this section, we connect CFM to the diffusion SBP with zero drift. This motivates the use of a topological drift, which we justify via spectral analysis as a topology-aware smoothness bias.

## 3.1 FLOW MATCHING SOLVES A DEGENERATE SCHRÖDINGER BRIDGE PROBLEM

CFM arises as a way of solving the SBP with trivial drift $b = 0$ and constant noise $\sigma \in \mathbb{R}_+$, in the limit $\sigma \to 0$. It is instructive, as a blueprint for TFM, to see this connection in terms of the three key CFM components.

**1. Conditional vector field $u_t^{x_0, x_1}$.** If $b = 0$ and $\sigma \in \mathbb{R}_+$, the diffusion bridge SDE simplifies to

$$\mathrm{d}X_t = u_t^{x_0, x_1}(X_t)\,\mathrm{d}t + \sigma\,\mathrm{d}W_t, \quad u_t^{x_0, x_1}(x) = \frac{x_1 - x}{1 - t}, \quad X_0 = x_0. \tag{9}$$

As $\sigma \to 0$, the dynamics reduce to the ODE $\dot{X}_t = u_t^{x_0, x_1}(X_t)$. Its unique solution is the straight line $X_t = (1 - t)x_0 + tx_1$. Thus, $u_t^{x_0, x_1}(X_t) = x_1 - x_0$, which recovers the CFM conditional vector field.

**2. Conditional path $\mathbb{P}_t^{x_0, x_1}$.** In the zero-noise limit, $(X_t \mid X_0 = x_0, X_1 = x_1)$ becomes deterministic. Thus, if $b = 0$, its law $\mathbb{P}_t^{x_0, x_1}$ converges to the CFM conditional law $\delta_{(1-t)x_0 + tx_1}$.

**3. Coupling $(X_0, X_1) \sim \pi$.** If $b = 0$ and $\sigma \in \mathbb{R}_+$, the entropic OT problem in Equation (5) simplifies to

$$\min_{(X_0, X_1) \sim \mathbb{Q}_{01}} \mathbb{E}\left[ \|X_1 - X_0\|^2 \right] + \sigma^2 D_{\mathrm{KL}}(\mathbb{Q}_{01} \| \mu_0 \otimes \mu_1), \quad \text{s.t. } \mathbb{Q}_{01} \in \Pi(\mu_0, \mu_1). \tag{10}$$

As $\sigma \to 0$, this converges to the exact OT problem in Equation (3) (Léonard, 2014). Therefore, $\mathbb{Q}_{01}^*$ coincides with the coupling $\pi^*$ used in OT-CFM. I-CFM can be seen as the independent approximation $\mathbb{Q}_{01}^* \approx \mu_0 \otimes \mu_1$.

Proposition 1 now shows that the marginal vector field $u$ of OT-CFM is the drift of the SBP solution

$$\mathrm{d}X_t = u_t(X_t)\,\mathrm{d}t + \sigma\,\mathrm{d}W_t, \quad X_0 \sim \mu_0.$$

In the limit $\sigma \to 0$, these dynamics converge to the flow ODE in Equation (1). Thus, OT-CFM can be viewed as solving the zero-noise limit of the diffusion Schrödinger bridge problem by learning the drift of this limiting ODE. This is a formal viewpoint, since an SBP with $\sigma = 0$ is generally not well-posed[3]. However, the optimal drift of the SBP does converge under $\sigma \to 0$, solving the *Benamou–Brenier OT problem*

$$\min \int_0^1 \tfrac{1}{2}\|u_t(x)\|^2 \mathbb{P}_t(\mathrm{d}x)\,\mathrm{d}t, \quad \text{s.t. } \partial_t \mathbb{P}_t + \nabla \cdot (\mathbb{P}_t u_t) = 0, \quad \mathbb{P}_0 = \mu_0, \quad \mathbb{P}_1 = \mu_1.$$

Intuitively, this drift is the minimum-energy vector field transporting $\mu_0$ to $\mu_1$ in Euclidean space.

## 3.2 TOPOLOGICAL REFERENCE PROCESS

**Topological diffusion.** CFM can be seen as solving the zero-noise limit of a drift-free SBP. Since the reference process plays the role of a prior, we can bias the SBP solution to respect topology of a simplicial complex $K$, by augmenting it with a topology-aware drift (Yang, 2025):

$$b_t(X_t) = H_t(\boldsymbol{L}_k)X_t + \alpha_t, \tag{11}$$

for a polynomial $H_t$—a choice achieving tractability and flexibility, as $H_t(\boldsymbol{L}_k)$ can approximate any analytic function of $\boldsymbol{L}_k$ by the Cayley–Hamilton theorem (Yang, 2025). The resulting *topological reference process*

$$\mathrm{d}X_t = H_t(\boldsymbol{L}_k)X_t + \alpha_t + \sigma\,\mathrm{d}W_t$$

serves as the starting point for derivation of TFM in Section 4.

---

[3]For instance, if $b = 0$, $\sigma = 0$, and $\mu_0 = \delta_{x_0}$, $\mathbb{P}$ is the Dirac delta on the constant path $t \mapsto x_0$. Any solution $\mathbb{Q}$ must satisfy $\mathbb{Q} \ll \mathbb{P}$, which since $\mathbb{P}$ is a point mass, implies $\mathbb{Q} = \mathbb{P}$ and, thus, $\mathbb{Q}_1 = \mathbb{P}_1 = \delta_{\omega_{x_0}}$. Therefore, unless $\mu_1 = \delta_{x_0}$, $\mathbb{Q}$ cannot satisfy the necessary condition $\mathbb{Q}_1 = \mu_1$. Consequently, the SBP has no solution.

Table 1: Flow ODE, conditional vector fields, and bridge process of CFM and TFM.

| Model | Flow ODE | $u_t^{x_0,x_1}(X_t)$ | $(X_t \mid X_0 = x_0, X_1 = x_1)$ |
|---|---|---|---|
| CFM | $\dot{X}_t = u_t(X_t)$ | $x_1 - x_0$ | $(1-t)x_0 + tx_1$ |
| TFM | $\dot{X}_t = -\kappa \boldsymbol{L}_k X_t + u_t(X_t)$ | $\Phi_{t,1}\tilde{\Sigma}_{1,1}^{-1}(x_1 - m_1(x_0))$ | $m_t(x_0) + \tilde{\Sigma}_{t,1}\tilde{\Sigma}_{1,1}^{-1}(x_1 - m_1(x_0))$ |

To motivate this choice, we focus on the case $H_t(\boldsymbol{L}_k) = -\kappa \boldsymbol{L}_k$ for $\kappa > 0$, as it connects to the heat equation and the heat kernel, providing a clear interpretation; however, other choices like $H_t(\boldsymbol{L}_k) = (\kappa^2 - \boldsymbol{L}_k)^{\nu+n_k/2}$ for $\kappa > 0$, $\nu > 1/2$ could be useful for their connection to the Matérn kernel (Borovitskiy et al., 2021). In the zero-noise limit the reference process becomes the heat equation $\dot{X}_t = -\kappa \boldsymbol{L}_k X_t$ with *diffusion rate* $\kappa$. Furthermore, let $\boldsymbol{U}_k = (u_1, \ldots, u_{n_k})$ be the eigenvectors and $\boldsymbol{D}_k = \mathrm{diag}(\lambda_0, \ldots, \lambda_{n_k})$ the eigenvalues of $\boldsymbol{L}_k$, so that $\boldsymbol{L}_k = \boldsymbol{U}_k \boldsymbol{D}_k \boldsymbol{U}_k^\top$. Under the change of coordinates $Y := \boldsymbol{U}_k^\top X$, the heat equation is diagonalized

$$\dot{X}_t = -\kappa \boldsymbol{L}_k X_t \iff \dot{Y}_t^i = -\kappa \lambda_i Y_t^i, \quad \forall i \in \{0, \ldots, n_k\}.$$

Its solution is given by $Y_t^i = \exp(-\kappa \lambda_i t) Y_0^i$. Thus, the eigenfunctions with non-zero eigenvalues decay exponentially quickly at a rate proportional to their frequency, while the eigenfunctions with zero eigenvalues, corresponding to topological features (cf. Section 2), stay constant. Therefore, our proposed drift can be seen as a bias dampening high-frequency oscillations—thereby denoising the signal—while preserving signal components aligned with the structural features of $K$, where bias strength is proportional to $\kappa$.

**Topological initial distribution.** For generative tasks, we can also inject topological information in the initial distribution $\mu_0$, by setting it to the heat Gaussian process $\mathcal{N}(0, \exp(-\kappa \boldsymbol{L}_k))$. Setting $\kappa = 0$ recovers the standard Gaussian—disregarding the structure of $K$ by disallowing heat flow between adjacent simplices.

## 4 TOPOLOGICAL FLOW MATCHING

CFM operates in Euclidean space, overlooking the topology of structured domains like graphs and simplicial complexes. Since OT-CFM learns the solution of the degenerate drift-free SBP, we propose to make it topology-aware, by augmenting the reference process with the topological drift from Equation (11). Retracing the derivation of OT-CFM from SBP, this time for the topological SBP

$$\mathrm{d}X_t = [H_t(\boldsymbol{L}_k)X_t + \alpha_t]\,\mathrm{d}t + \sigma\,\mathrm{d}W_t, \quad X_0 \sim \mu_0,$$

yields *topological flow matching* (TFM)—a principled, topology-aware extension of FM. Although inspired by topological Schrödinger bridge matching (TSBM) (Yang, 2025), in stark contrast to it, TFM enjoys the key advantages of standard FM: scalable, simulation-free training and deterministic sample paths.

To see why the SBP perspective is useful, let us notice that, if we augment the flow ODE with the topological drift, yielding the *topological flow ODE*

$$\dot{X}_t = H_t(\boldsymbol{L}_k)X_t + \alpha_t + u_t(X_t), \tag{12}$$

a sensible choice of $u_t^{x_0,x_1}(X_t)$ is not clear. For instance, $u_t^{x_0,x_1}(X_t) = (x_1 - x_0) - (H_t(\boldsymbol{L}_k)X_t + \alpha_t)$ yields the same conditional path as CFM, which ignores the topology:

$$\dot{X}_t = (H_t(\boldsymbol{L}_k)X_t + \alpha_t) + u_t^{x_0,x_1}(X_t) = x_1 - x_0. \tag{13}$$

The degenerate SBP, however, has a unique solution, which induces a principled choice of $u_t^{x_0,x_1}(X_t)$—as stated in Proposition 2.

As in the Euclidean case, while the zero-noise limit of the topological SBP is formal, its optimal drift $u_t$ converges in the limit $\sigma \to 0$ to the minimizer of the dynamic optimal control problem

$$\min \int_0^1 \tfrac{1}{2}\|u_t(x)\|^2 \mathbb{P}_t(\mathrm{d}x)\,\mathrm{d}t, \quad \text{s.t. } \partial_t \mathbb{P}_t + \nabla\cdot(\mathbb{P}_t(H_t(\boldsymbol{L}_k)+\alpha_t+u_t)) = 0, \quad \mathbb{P}_0 = \mu_0, \quad \mathbb{P}_1 = \mu_1.$$

Intuitively, this $u_t$ is the minimum-energy corrective force needed to steer from $\mu_0$ to $\mu_1$ when the system, left undisturbed, evolves according to the heat equation.

**Notation.** We denote $m_{t|s}(x) := \mathbb{E}[X_t \mid X_s = x]$, $\tilde{\Sigma}_{s,t|r} := \sigma^{-2}\mathrm{Cov}(X_s, X_t \mid X_r = x)$, with the shorthands $m_t := m_{t|0}$, $\tilde{\Sigma}_{s,t} := \tilde{\Sigma}_{s,t|0}$. We write $\Phi_{s,t}$ for the solution to $\dot{\Phi}_{s,t} = H_t(\boldsymbol{L}_k)\Phi_{s,t}$, $\Phi_{s,s} = I_{n_k}$.

---

**Proposition 2** (Conditional control $u_t^{x_0,x_1}$). *Let $X_t$ be the topological reference process. The bridge $(X_t \mid X_0 = x_0, X_1 = x_1)$ follows the SDE $\mathrm{d}X_t = [H_t(\boldsymbol{L}_k)X_t + \alpha_t + u_t^{x_0,x_1}(X_t)]\,\mathrm{d}t + \sigma\,\mathrm{d}W_t$, with*

$$u_t^{x_0,x_1}(X_t) = \Phi_{t,1}\tilde{\Sigma}_{1|t}^{-1}(x_1 - m_{1|t}(X_t)). \tag{14}$$

*In the limit $\sigma \to 0$, $u_t^{x_0,x_1}(X_t)$ becomes independent of $X_t$: $u_t^{x_0,x_1}(X_t) = \Phi_{t,1}\tilde{\Sigma}_{1,1}^{-1}(x_1 - m_1(x_0))$. For $H_t(\boldsymbol{L}_k) = -\kappa\boldsymbol{L}_k$, we have the simple formulas in spectral coordinates*

$$(u_t^{y_0,y_1}(Y_t))^i = \begin{cases} y_1^i - y_0^i & \lambda^i = 0 \text{ or } \kappa = 0 \\ \frac{2\kappa\lambda^i \exp(-\kappa\lambda^i(1-t))}{\exp(2\kappa\lambda^i)-1}(y_1^i - \exp(-\kappa\lambda^i)y_0^i) & \text{else.} \end{cases}$$

---

TFM learns the topological flow ODE in the same way as CFM learns the standard flow ODE: by choosing a coupling $(X_0, X_1) \sim \pi$ and minimizing the CFM loss

$$\mathbb{E}_{t\sim\mathrm{Unif}[0,1),\,(X_0,X_1)\sim\pi,\,X\sim\mathbb{P}_t^{X_0,X_1}}\left[\|u_t^{X_0,X_1}(X) - u_t^\theta(X)\|^2\right].$$

This is a simulation-free objective, since the bridge $(X_t \mid X_0 = x_0, X_1 = x_1)$ is deterministic.

---

**Proposition 3** (Conditional path $\mathbb{P}_t^{x_0,x_1}$). *Let $X_t$ be the topological reference process. The bridge $(X_t \mid X_0 = x_0, X_1 = x_1)$ has the mean*

$$m_t^{x_0,x_1} = m_t(x_0) + \tilde{\Sigma}_{t,1}\tilde{\Sigma}_{1,1}^{-1}(x_1 - m_1(x_0)).$$

*As $\sigma \to 0$, the bridge law concentrates on the mean $\mathbb{P}_t^{x_0,x_1} = \delta_{m_t^{x_0,x_1}}$. For $H_t(\boldsymbol{L}_k) = -\kappa\boldsymbol{L}_k$, we have*

$$(m_t^{y_0,y_1})^i = \begin{cases} ty_1^i + (1-t)y_0^i & \kappa = 0 \text{ or } \lambda^i = 0 \\ \frac{\sinh(\kappa\lambda^i(1-t))}{\sinh(\kappa\lambda^i)}y_0^i + \frac{\sinh(\kappa\lambda^i t)}{\sinh(\kappa\lambda^i)}y_1^i & \text{else.} \end{cases}$$

---

Any coupling $\pi$ yields a valid TFM variant. However, only the *OT-TFM* variant with $\pi = \mathbb{Q}_{01}^*$ solves the degenerate topological SBP. Fortunately, $\mathbb{Q}_{01}^*$ can be computed as efficiently as in the Euclidean case, since the transport cost $c$ corresponding to the topological SBP has a simple formula given in Proposition 4.

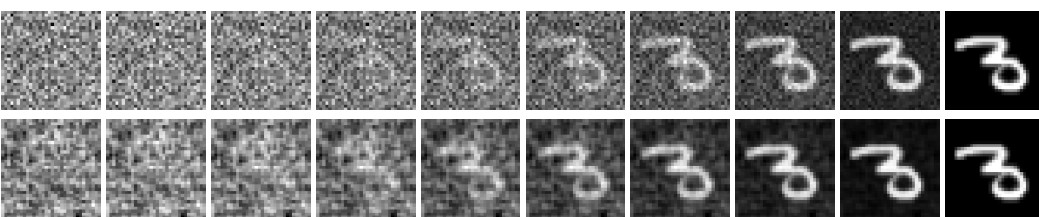

Figure 2: Conditional paths from noise to data on the MNIST dataset at $t$ intervals of $1/9$, showing the smoothing effect of the topological reference process. *Top:* CFM. *Bottom:* TFM $\kappa = 0.5, \mu_0 = \mathcal{N}(0, -\kappa\boldsymbol{L}_k)$.

Table 2: Mean 1-Wasserstein distance, $\pm$ 1 standard deviation, of CFM and TFM on real-world datasets from Yang (2025) compared against the best performing TSBM variant. We omit their GTSB result on ocean currents, since it assumes analytically known boundary distributions.

| Method | Earthquakes | Traffic flows | Brain fMRI | Single-cell | Ocean currents |
|--------|-------------|---------------|------------|-------------|----------------|
| I-CFM | $8.37_{\pm 0.05}$ | $1.72_{\pm 0.01}$ | $11.71_{\pm 0.02}$ | $0.022_{\pm 0.001}$ | $1.95_{\pm 0.02}$ |
| OT-CFM | $8.25_{\pm 0.06}$ | $1.59_{\pm 0.01}$ | $11.30_{\pm 0.01}$ | $0.019_{\pm 0.001}$ | $2.00_{\pm 0.05}$ |
| I-TFM | $\mathbf{4.93}_{\pm \mathbf{0.06}}$ | $\mathbf{1.27}_{\pm \mathbf{0.01}}$ | $6.33_{\pm 0.02}$ | $\mathbf{0.018}_{\pm \mathbf{0.001}}$ | $\mathbf{1.87}_{\pm \mathbf{0.04}}$ |
| OT-TFM | $5.53_{\pm 0.02}$ | $\mathbf{1.27}_{\pm \mathbf{0.00}}$ | $\mathbf{5.86}_{\pm \mathbf{0.01}}$ | $0.019_{\pm 0.001}$ | $1.91_{\pm 0.06}$ |
| TSBM (best) | $7.69_{\pm 0.04}$ | $9.92_{\pm 0.02}$ | $7.51_{\pm 0.01}$ | $0.140_{\pm 0.010}$ | $6.89_{\pm 0.00}$ |

**Proposition 4** (Coupling $(X_0, X_1) \sim \mathbb{Q}_{01}^*$)**.** *For the topological reference process $X$, the entropic OT problem in Equation (5) is equivalent to*

$$\min_{(X_0, X_1) \sim \mathbb{Q}_{01}} \mathbb{E}[c] + \sigma^2 D_{KL}(\mathbb{Q}_{01} \| \mu_0 \otimes \mu_1), \quad s.t. \; \mathbb{Q}_{01} \in \Pi(\mu_0, \mu_1), \tag{15}$$

*and, in the limit $\sigma \to 0$, converges to an exact OT problem with the same transport cost $c$, given by*

$$c(x_0, x_1) = (x_1 - m_1(x_0))^\top \tilde{\Sigma}_{1,1}^{-1} (x_1 - m_1(x_0)).$$

*With $H_t(\boldsymbol{L}_k) = -\kappa \boldsymbol{L}_k$, we have the spectral formula $c(y_0, y_1) = \sum_{i=1}^{n_k} c_i(y_0^i, y_1^i)$, with*

$$c_i(y_0^i, y_1^i) = \begin{cases} (y_1^i - y_0^i)^2 & \lambda^i = 0 \text{ or } \kappa = 0 \\ \frac{2\kappa\lambda^i}{1-\exp(-2\kappa\lambda^i)}(y_1^i - \exp(-\kappa\lambda^i)y_0^i)^2 & else. \end{cases} \tag{16}$$

Thus, TFM enjoys scalable, simulation-free formulas mirroring CFM, summarized in Table 1. Any variant of TFM, such as *I-TFM* with $\pi = \mu_0 \otimes \mu_1$, exploits topological information via the flow ODE, the conditional vector fields, and the conditional path, while OT-TFM also uses it in the coupling.

## 5 EXPERIMENTS

We developed TFM to improve performance over CFM on generative tasks over topological signals by exploiting the structure of their domains. To examine whether this goal has been achieved, we compare TFM—specifically its heat flow variant with the topological drift $-\kappa \boldsymbol{L}_k X_t$—against CFM on diverse real-world datasets on graphs and 2-simplicial complexes, compiled by Yang (2025) to evaluate TSBM. This also lets us compare TFM against TSBM, showing whether TFM can serve as a scalable, simulation-free alternative. To ensure a fair comparison, we use the data, models, and experimental setup given by Yang (2025) wherever possible. We use a fixed $\kappa = 2.0$, though a choice tailored to a given experiment can improve performance (cf. Section F.3). We also investigate if the smoothing effect TFM has on conditional paths (cf. Figure 2) can aid image generation, by viewing images as node signals on the grid, using $\kappa = 0.01$. The experiments are divided into: *generation*, where the initial distribution is simple and the final distribution is a data distribution; and *matching*, where the initial and final distributions are different data distributions.

Table 3: Summary of topological datasets. $k$: signal order (0=node, 1=edge). $N$: number of data points, $\infty$ if synthetic.

| | Earthquake | Traffic | fMRI | Cell | Ocean |
|------|------------|---------|------|------|-------|
| Task | Generation | Generation | Matching | Matching | Matching |
| $K$ | Graph | Mesh | Graph | Graph | Mesh |
| $k$ | 0 | 1 | 0 | 0 | 1 |
| $n_k$ | 576 | 340 | 360 | 18K | 20K |
| $N$ | 28 | 17K | 2K | 6K | $\infty$ |

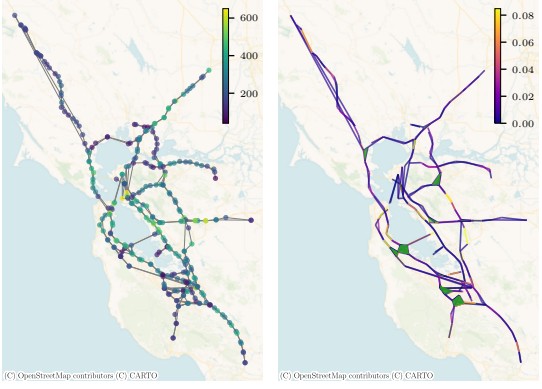 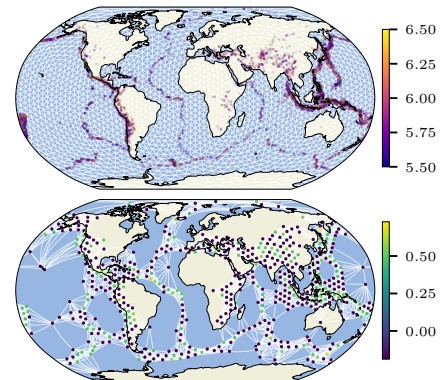

Figure 3: *Left:* Road network in the traffic flow experiment with a node signal. *Right:* Simplicial complex built from the road network, with triangles shown in green, and an edge signal.

Figure 4: *Top:* Historical earthquake events overlaid with a mesh discretizing the globe. *Bottom:* Graph used in the earthquake experiment with an example node signal.

**Generation.** We first tackle modeling of earthquake magnitudes around the globe, using data between 1990 and 2017 from the IRIS dataset. To capture regions of seismic activity, we discretize the Earth's surface as a mesh and construct a 10-nearest-neighbors graph from the nodes closest to earthquake events (cf. Figure 4). For each year, events are mapped to nodes and their magnitudes are averaged to create a target signal. Secondly, we consider traffic flow modeling using the PeMSD4 dataset, which contains node signals from 307 measuring stations. Following Chen et al. (2022), signals are lifted to the edges and the road network graph to a 2-simplicial complex (cf. Figure 3). From Table 2, we find that TFM significantly outperforms CFM and TSBM. Interestingly, OT-TFM is outperformed by I-TFM on the earthquake experiment.

We also test image generation on the CIFAR-10 dataset, treating images as node signals on a 32-by-32-by-3 (width, height, channels) grid. From Table 4, we see that I-TFM and OT-TFM achieve a small mean and median improvement over I-CFM and OT-CFM respectively, with a relatively high variance between runs. The advantage of TFM is much larger on the earthquake and traffic experiments, where the signal domains have complex structure, than in image generation, where the domain is a regular grid. This suggests that, while TFM shows some promise for image generation, it gains significant advantage by capturing topological features of complex domains.

Table 4: The mean, median, and standard deviation of FID scores on CIFAR-10 generation over 10 runs.

| Method | Mean | Median | SD |
|--------|--------|--------|--------|
| I-CFM | 3.7005 | 3.7061 | 0.0462 |
| OT-CFM | 3.8238 | 3.8308 | 0.0615 |
| I-TFM | 3.6972 | 3.6795 | 0.0821 |
| OT-TFM | 3.8107 | 3.8046 | 0.0771 |

**Matching.** We first consider edge signals representing water currents on a 2-simplicial complex discretizing the North Atlantic Ocean, extracted by Chen & Meila (2021) from data collected by NOAA Atlantic Oceanographic and Meteorological Laboratory. As the initial distribution we use an edge GP fitted to data (Yang et al., 2024), and for the final distribution we choose a synthetic curl-free Gaussian process. The target samples exhibit only sinks and sources, while the initial samples form realistic patterns (cf. Figure 6). Secondly, we study the differentiation of cells in an embryoid body across 5 time steps, using data from Moon et al. (2019); Tong et al. (2020). Using data from every time step, we construct a 4-nearest-neighbors graph capturing the structure of the temporal cell evolution. Then, as the initial and final distributions we use a normalized indicator function over the data at the first and last timestep. Lastly, we consider the graph of 360 functional regions of the brain, with edges weighted by connection strength. We use 1,190 fMRI signals from the Human Connectome Project, decomposed into time-fluctuating *aligned* and time-persistent *liberal* components, serving as initial and final distributions respectively. From Table 2, we see that TFM outperforms CFM on all experiments—most of all on the brain graph. TFM also consistently outperforms TSBM, demonstrating the advantage of a simulation-free framework.

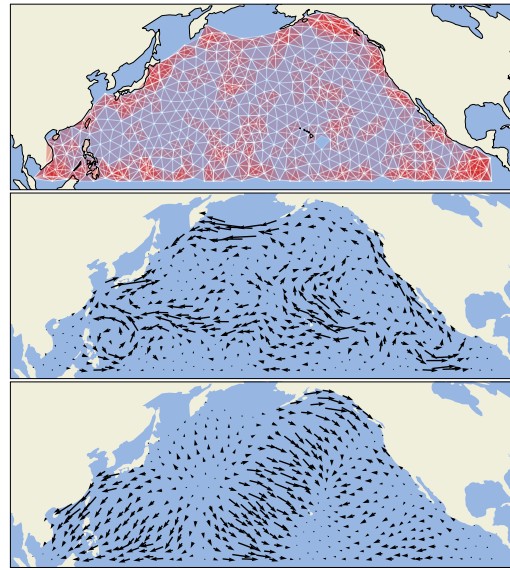

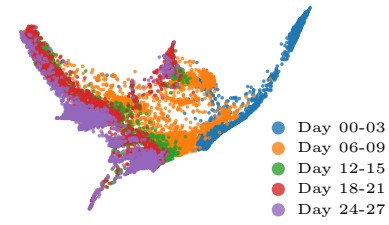

Figure 5: Single-cell data in the PHATE co-ordinates.

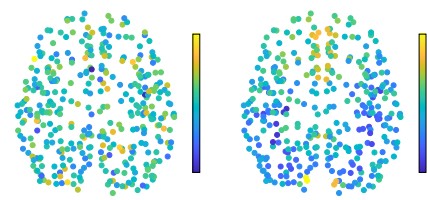

Figure 6: Ocean experiment data. *Top:* Ocean mesh (subset). *Middle:* Initial sample. *Bottom:* Final sample.

Figure 7: Initial (left) and final (right) brain fMRI signals.

## 6 CONCLUSION

We introduced topological flow matching, a topology-aware generalization of flow matching for generative modeling over signals on structured spaces. We proposed a way to incorporate topological information into FM by utilizing its relation to the Schrödinger bridge problem and augmenting the reference process with a Laplacian-derived drift. This addition can be seen as a prior which smooths signal components that do not correspond to topological features captured by the Laplacian kernel. Furthermore, we proved that TFM retains a stable, scalable, and simulation-free objective as well as deterministic sample paths, which makes it a drop-in replacement for standard FM. We demonstrated performance improvements over FM and topological Schrödinger bridge matching on a variety of datasets on graphs and simplicial complexes, with the greatest advantage on the most complex structures. While we focused on the natural choice of topological drift based on the heat equation, future work could further explore different reference processes, including Matérn-like drifts or drifts with a learned diffusion rate $\kappa$. In addition, a drift with a time-dependent Laplacian could enable matching tasks between signals on two different spaces. Finally, variants of TFM for signals on manifolds could be examined for applications e.g. in geostatistics and robotics. Altogether, TFM provides a principled extension of FM that improves performance on structured spaces and opens new avenues for both theoretical exploration and practical applications.

## ACKNOWLEDGEMENTS

KW thanks Maosheng Yang for providing additional theoretical and experimental details of (Yang, 2025).

## ETHICS STATEMENT

Our work is primarily focused on theoretical algorithmic development for faster and more accurate generative models for topological datatypes, with reduced focus on experimental implementation. However, we recommend that future users of our work exercise appropriate caution when applying it to domains that may involve sensitive considerations.

## Reproducibility Statement

To ensure the reproducibility of our work, we provide detailed information regarding our experimental setup, theoretical claims and datasets. An implementation of topological flow matching and the code to reproduce all experiments will be made available upon publication. The theoretical claims and derivations underlying TFM are presented in Section A. All datasets used in our experiments are publicly available at their cited source in Section 5. Further implementation details are available in Section E.

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

APPENDIX

# A  PROOFS

## A.1  PROOF OF PROPOSITION 1

This proposition is a direct corollary of Proposition 1 of generator matching (GM) (Holderrieth et al., 2025), which states that for any Markov process $X$ with law $\mathbb{P}$, the generator $\mathcal{L}_t$ of $X$ is given as the mixture of generators $\mathcal{L}_t^{x_1}$ of $(X \mid X_1 = x_1)$

$$\mathcal{L}_t f(x) = \mathop{\mathbb{E}}_{X_1 \sim \mathbb{P}_1(\cdot \mid X_t = x)} [\mathcal{L}_t^{X_1} f(x)]. \tag{17}$$

Now, a general diffusion SDE

$$\mathrm{d}X_t = b_t(X_t)\,\mathrm{d}t + \sigma_t(X_t)\,\mathrm{d}W_t \tag{18}$$

has the generator

$$\mathcal{L}_t f(x) = \langle b_t(x), \nabla f(x) \rangle + \langle \tfrac{1}{2} a_t(x), \nabla^2 f(x) \rangle,$$

where $a_t(x) := \sigma_t(x)\sigma_t^\top(x)$, the first inner product is the vector inner product, and the second inner product is the Frobenius matrix inner product. It follows that the bridge $X \mid X_0 = x_0, X_1 = x_1$ with the dynamics

$$\mathrm{d}X_t = [b_t(X_t) + u_t^{x_0,x_1}(X_t)]\,\mathrm{d}t + \sigma_t(X_t)\,\mathrm{d}W_t, \tag{19}$$

and $u_t^{x_0,x_1}(x) = a_t(x)\nabla \log p_{t,1}(x, x_1)$, has the generator

$$\mathcal{L}_t^{x_1} f(x) = \langle b_t(x) + u_t^{x_0,x_1}(x), \nabla f(x) \rangle + \langle \tfrac{1}{2} a_t(x), \nabla^2 f(x) \rangle.$$

Moreover, noting the SBP solution by $X^* \sim \mathbb{Q}^*$, we have that $\mathcal{L}^{x_1}$ is also the generator of $X^* \mid X_0^* = x_0, X_1^* = x_1$, because $(\mathbb{Q}^*)^{x_0,x_1} = \mathbb{P}^{x_0,x_1}$. Additionally, since the generator does not depend on the initial distribution, $\mathcal{L}_t^{x_1}$ is also the generator of $X^* \mid X_1^* = x_1$. Taking expectations, it follows from Equation (17) that the SBP solution has the generator

$$\mathcal{L}_t^* f(x) = \left\langle b_t(x) + \mathop{\mathbb{E}}_{X_1^* \sim \mathbb{Q}_1^*(\cdot \mid X_t^* = x)}\left[u_t(x)^{X_0^*, X_1^*}\right], \nabla f(x) \right\rangle + \langle \tfrac{1}{2} a_t(x), \nabla^2 f(x) \rangle,$$

and so follows the SDE

$$\mathrm{d}X_t^* = [b_t(X_t^*) + u_t(X_t^*)]\,\mathrm{d}t + \sigma_t(X_t^*)\,\mathrm{d}W_t.$$

with

$$u_t(X_t) := \mathop{\mathbb{E}}_{X_1^* \sim \mathbb{Q}_1^*(\cdot \mid X_t^*)}\left[u_t^{X_0^*, X_1^*}(X_t^*)\right].$$

Since neither side of the equation actually depends on $X_0$, we can take expectation over $\mathbb{Q}_0^*(\cdot \mid X_t^*, X_1^*)$ obtaining

$$u_t(X_t) = \mathop{\mathbb{E}}_{X_1^* \sim \mathbb{Q}_1^*(\cdot \mid X_t^*),\ X_0^* \sim \mathbb{Q}_0^*(\cdot \mid X_t^*, X_1^*)}\left[u_t^{X_0^*, X_1^*}(X_t^*)\right] = \mathop{\mathbb{E}}_{(X_0^*, X_1^*) \sim \mathbb{Q}_{01}^*(\cdot \mid X_t^*)}\left[u_t^{X_0^*, X_1^*}(X_t^*)\right].$$

Renaming $X^*$ to $X$ finishes the proof.

## A.2  PROOF OF PROPOSITIONS 2 AND 3

**General topological reference process.**  To simplify notation, let $A_t = H_t(\boldsymbol{L}_k)$, so that the topological reference process takes the form

$$\mathrm{d}X_t = (A_t X_t + \alpha_t)\,\mathrm{d}t + \sigma\,\mathrm{d}W_t. \tag{20}$$

This is an example of a *linear Gaussian* SDE. It is well-known that a linear Gaussian SDE can be expressed for any $0 \le r < t$ as

$$X_t = \Phi_{r,t} X_r + \int_r^t \Phi_{s,t}\alpha_s\,\mathrm{d}s + \sigma \int_r^t \Phi_{s,t}\,\mathrm{d}W_s,$$

where $\Phi_{s,t}$ is the solution of the ODE

$$\dot{\Phi}_{s,t} = A_t\Phi_{s,t}, \quad \Phi_{s,s} = I_{n_k}.$$

It follows that the conditional process $(X_t \mid X_r = x)$ is a Gaussian process with the mean $m_{t|r}(x)$ and covariance $\Sigma_{s,t|r}$

$$m_{t|r}(x) := \mathbb{E}[X_t \mid X_r = x_r] = \Phi_{r,t}x_r + \int_r^t \Phi_{s,t}\alpha_s \, ds,$$

$$\Sigma_{s,t|r} := \sigma^2\tilde{\Sigma}_{s,t|r},$$

where

$$\tilde{\Sigma}_{s,t|r} := \int_r^{s\wedge t} \Phi_{u,t}\Phi_{u,s}^\top \, du.$$

This yields Gaussian transition probabilities

$$p_{t,1}(x,x_1) = \mathcal{N}(x_1; \mathbb{E}[X_1 \mid X_t = x_t], \mathrm{Var}(X_1 \mid X_t = x_t)) = \mathcal{N}(x_1; m_{1|t}(x), \Sigma_{1,1|t}).$$

Thus, the conditional control $u_t^{x_0,x_1}$ of the corresponding bridge SDE takes the form

$$u_t^{x_0,x_1}(X_t) = -\sigma^2\nabla_x \log \mathcal{N}(x; m_{1|t}(x), \Sigma_{1,1|t})$$

$$= \sigma^2\Phi_{t,1}\Sigma_{1,1|t}^{-1}(x_1 - m_{1|t}(X_t))$$

$$= \Phi_{t,1}\tilde{\Sigma}_{1,1|t}^{-1}(x_1 - m_{1|t}(X_t))$$

Moreover, conditional Gaussian process formulas show that the $(X_t \mid X_0 = x_0, X_1 = x_1)$ is also a Gaussian process with the mean $m_t^{x_0,x_1}$ and covariance $\Sigma_{s,t}^{x_0,x_1}$:

$$m_t^{x_0,x_1} := \mathbb{E}[X_t \mid X_0 = x_0, X_1 = x_1] = m_t(x_0) + \Sigma_{t,1}\Sigma_{1,1}^{-1}(x_1 - m_1(x_0))$$

$$= m_t(x_0) + \tilde{\Sigma}_{t,1}\tilde{\Sigma}_{1,1}^{-1}(x_1 - m_1(x_0)),$$

$$\Sigma_{s,t}^{x_0,x_1} := \mathrm{Cov}(X_s, X_t \mid X_0 = x_0, X_1 = x_1)$$

$$= \Sigma_{s,t} - \Sigma_{s,1}\Sigma_{1,1}^{-1}\Sigma_{1,t}$$

$$= \sigma^2\left(\tilde{\Sigma}_{s,t} - \tilde{\Sigma}_{s,1}\tilde{\Sigma}_{1,1}^{-1}\tilde{\Sigma}_{1,t}\right),$$

where $m_t := m_{t|0}$ and $\Sigma_{s,t} := \Sigma_{s,t|0}$. Taking the limit $\sigma \to 0$, these $u_t^{x_0,x_1}$ and $m_t^{x_0,x_1}$ stay constant, while the law of the bridge $(X_t \mid X_0 = x_0, X_1 = x_1)$ converges to

$$(X_t \mid X_0 = x_0, X_1 = x_1) \sim \mathbb{P}_t^{x_0,x_1} = \delta_{m_t^{x_0,x_1}}.$$

Finally, we can use the fact that $X_t \mid X_0 = x_0, X_1 = x_1$ is deterministic, to further simplify $u^{x_0,x_1}$

$$u_t^{x_0,x_1}(X_t) = \Phi_{t,1}\tilde{\Sigma}_{1|t}^{-1}(x_1 - m_{1|t}(m_t^{x_0,x_1})) = \Phi_{t,1}\tilde{\Sigma}_{1,1}^{-1}(x_1 - m_1(x_0)),$$

yielding a stable conditional control independent of $X_t$. This proves the general formulas in Propositions 2 and 3.

**Heat topological reference process.** The spectral coordinates $Y := U_k^\top X$ diagonalize the topological reference process and hence the formulas for $u_t^{x_0,x_1}$ and $(X_t \mid X_0 = x_0, X_1 = x_1)$, which we denote $u_t^{y_0,y_1}$ and $Y_t \mid Y_0 = y_0, Y_1 = y_1$. It follows then by simple algebraic manipulation that for $H_t(\lambda^i) = -\kappa\lambda^i$, we have the spectral formulas:

$$\Phi_{s,t}^i = \exp(-\kappa\lambda^i(t-s)),$$

$$m_1(y)^i = \exp(-\kappa\lambda^i)y^i,$$

$$\tilde{\Sigma}_{1,1}^{i,i} = \int_0^1 \exp(-2\kappa\lambda^i(1-t)) \, dt = \begin{cases} 1 & \kappa = 0 \text{ or } \lambda^i = 0 \\ \frac{1-\exp(-2\kappa\lambda^i)}{2\kappa\lambda^i} & \text{else,} \end{cases}$$

$$(u_t^{y_0,y_1}(Y_t))^i = \begin{cases} y_1^i - y_0^i & \kappa = 0 \text{ or } \lambda^i = 0 \\ \frac{2\kappa\lambda^i\exp(-\kappa\lambda^i(1-t))}{1-\exp(-2\kappa\lambda^i)}(y_1^i - \exp(-\kappa\lambda^i)y_0^i) & \text{else,} \end{cases}$$

$$(m_t^{y_0,y_1})^i = \begin{cases} ty_1^i + (1-t)y_0^i & \kappa = 0 \text{ or } \lambda^i = 0 \\ \frac{\sinh(\kappa\lambda^i(1-t))}{\sinh(\kappa\lambda^i)}y_0^i + \frac{\sinh(\kappa\lambda^i t)}{\sinh(\kappa\lambda^i)}y_1^i & \text{else} \end{cases},$$

(21)

which was to be shown.

### A.3 PROOF OF PROPOSITION 4.

We can use the formulas found in Section A.2 to find the transport cost $c = \log \frac{\mathrm{d}\mu_0 \otimes \mu_1}{\mathrm{d}\mathbb{P}_{01}}$ when $\mathbb{P}$ is the law of a process with the linear Gaussian SDE dynamics (cf. Equation (20))

**General topological reference process.** To this end, we first notice that if $\mu_1$ has Lebesgue density $\rho$, then the transport cost factorizes as

$$\log \frac{\mathrm{d}\mu_0 \otimes \mu_1}{\mathrm{d}\mathbb{P}_{01}}(x_0, x_1) = \log \frac{\mathrm{d}\mu_1}{\mathrm{d}\mathbb{P}_1(\cdot \mid X_0 = x_0)}(x_1) = \log \rho(x_1) - \log p_{0,1}(x_0, x_1). \quad (22)$$

Since $\mathbb{E}_{\mathbb{Q}_{01}}[\log \rho(X_1)] = \mathbb{E}_{\mu_1}[\log \rho(X_1)]$, the first summand does not change the minimizer of the associated entropic OT problem. Hence, in solving this problem, it is equivalent to consider the cost

$$c(x_0, x_1) = -\log p_{0,1}(x_0, x_1) = \log \mathcal{N}(x_1; m_1(x_0), \Sigma_{1,1}) \propto (x_1 - m_1(x_0))^\top \Sigma_{1,1}^{-1}(x_1 - m_1(x_0)), \quad (23)$$

where we drop the constant $\frac{1}{2}\log(2\pi\Sigma_{1,1})$, since it too does not affect the minimizer. Furthermore, we can extract out $\sigma$

$$c(x_0, x_1) = (x_1 - m_1(x_0))^\top \Sigma_{1,1}^{-1}(x_1 - m_1(x_0)) = \sigma^{-2}(x_1 - m_1(x_0))^\top \tilde{\Sigma}_{1,1}^{-1}(x_1 - m_1(x_0)) \quad (24)$$

Multiplying the associated entropic OT problem by $\sigma^2$, we get the equivalent problem

$$\min_{(X_0, X_1) \sim \mathbb{Q}_{01}} \mathbb{E}\left[(X_1 - m_1(X_0))^\top \Sigma_{1,1}^{-1}(X_1 - m_1(X_0))\right] + \sigma^2 D_{\mathrm{KL}}(\mathbb{Q}_{01} \| \mu_0 \otimes \mu_1), \qquad \text{s.t. } \mathbb{Q}_{01} \in \Pi(\mu_0, \mu_1). \quad (25)$$

Since the cost is now no longer dependent on $\sigma$, we can pass to the zero noise limit obtaining the exact OT problem

$$\min_{(X_0, X_1) \sim \mathbb{Q}_{01}} \mathbb{E}\left[(X_1 - m_1(X_0))^\top \Sigma_{1,1}^{-1}(X_1 - m_1(X_0))\right], \qquad \text{s.t. } \mathbb{Q}_{01} \in \Pi(\mu_0, \mu_1), \quad (26)$$

what was to be shown.

**Heat topological reference process.** In the spectral coordinates, the covariance of $X$ is diagonal; hence, the probability density $p_{0,1}(Y_0, Y_1)$ factorizes as $\prod_{i=1}^{n_k} p_{0,1}(Y_0^i, Y_1^i)$. Consequently, the cost factorizes $c(y_0, y_1) = \sum_{i=1}^{n_k} c_i(y_0^i, y_1^i)$ with

$$c_i(y_0^i, y_1^i) = (\Sigma_{1,1}^{-1})^{i,i}(y_1^i - m_1(y_0)^i)^2.$$

Finally, in the case $H_t(\lambda_i) = -\kappa\lambda^i$, plugging in the expression from Equation (21) simplifies the transport cost even further. Most simply, if either $\kappa = 0$ or $\boldsymbol{L}_k = 0$, we get

$$c_i(y_0^i, y_1^i) = \left(y_1^i - y_0^i\right)^2 / 1 = \left(y_1^i - y_0^i\right)^2. \quad (27)$$

Otherwise, we still get an efficient formula

$$c_i(y_0^i, y_1^i) = \frac{2\kappa\lambda^i}{1 - \exp(-2\kappa\lambda^i)}\left(y_1^i - \exp(-\kappa\lambda^i)y_0^i\right)^2, \quad (28)$$

which was to be proven.

## B ADDITIONAL RELATED WORK

### B.1 TOPOLOGICAL SCHRÖDINGER BRIDGE MATCHING

**Absence of a Topological Flow-Matching Framework in Yang (2025).** To the best of our knowledge, Yang (2025) does not present a topological flow matching framework. The closest relevant result is Corollary E.2, which provides the probability-flow ODE associated with the solution of the topological SBP. However, the drift in this ODE is expressed in terms of two auxiliary random variables defined via a coupled system of heat equations, making it computationally intractable and therefore unsuitable as a basis for flow matching.

**Distinction Between TFM and TSBM.** TFM differs fundamentally from TSBM along the same axis that distinguishes flow matching from Schrödinger bridge matching:

- **Simulation-free vs. simulation-based training.** TFM inherits the simulation-free training paradigm of FM: its objective requires only deterministic evaluations of the vector field, and no stochastic sample path simulation is involved. In contrast, TSBM inherits the simulation-based nature of SBM and relies on stochastic path sampling during training. This yields substantial empirical benefits for TFM: faster training, increased numerical stability, and more direct scalability to high-dimensional settings.
- **Deterministic vs. stochastic sample paths.** TFM produces deterministic sample paths governed by the learned flow ODE, while TSBM yields stochastic paths arising from diffusion processes. This is a qualitative difference intrinsic to the two formulations.

In summary, TFM retains the key properties of FM (scalable, simulation-free, deterministic), whereas TSBM retains those of SBM (simulation-based, stochastic). The methodological gap between TFM and TSBM is therefore as substantial as the gap between FM and SBM.

**Topology-Aware Initialisation.** For generative modelling, we additionally propose a topology-aware initial distribution, which can further improve fidelity of CFM and TFM. Empirical evidence for this design choice is provided in Section F.2.

### B.2 OTHER ARCHITECTURES AND FRAMEWORKS

In this work, we consider a topological form of flow matching which generates signals with respect to a specific topological space. A large literature exists on graph signal processing (Isufi et al., 2025) and geometric deep learning (Bronstein et al., 2021). In this work we use simple architectures, but note that TFM could potentially benefit from the vast literature on topological deep learning architectures (Yang et al., 2025; Battiloro et al., 2024; Goh et al., 2022), or graph neural networks (in the case of simple 1-simplices) (Kipf & Welling, 2017).

This literature is in contrast to that of generating topologies (Papamarkou et al., 2024), which is a related field with many potential synergies, but is not directly applicable to our setting.

## C SKETCH OF EXTENSIONS

We sketch how TFM extends to vector-valued signals, countably infinite simplicial complexes, and compact Riemannian manifolds. For the latter two cases, the guiding principle is that once the reference Brownian motion is defined in the correct infinite-dimensional space and mild regularity holds, the linear Gaussian reference dynamics are well posed, bridges are Gaussian, and the formulas from Section 4 carry over in the sense of operator calculus. However, since the Laplacian eigendecomposition is infinite, it must be finitely truncated to handle in practice. This eliminates most functional analytic technicalities, since the setting becomes finite-dimensional.

**Vector-valued signals.** To model distributions over vector-valued $k$-simplex signals $X_t \colon K_k \to \mathbb{R}^d$, we can simply assume that the topology acts only *spatially* and not across output dimensions. This simply means that we identify the signal $X_t$ with an element of $\mathbb{R}^{n_k d}$ and apply TFM with the block-diagonal Laplacian:
$$\boldsymbol{L}_k^{\text{vec}} := \boldsymbol{L}_k \otimes I_d.$$

**Signals on infinite simplicial complexes** Let $K$ be a countably infinite, but locally finite, simplicial complex and fix $k$. We can model $k$-simplex signals in the Hilbert space $\mathcal{H} = \ell^2(K_k)$. A direct sum of i.i.d. 1-dimensional Brownian motions does *not* produce $\ell^2$-valued paths: $\sum_i \|W_t^i e_i\|^2$ has infinite expected value, so paths "blow up" immediately. To remedy this, we can use a $Q$-*Brownian motion*, which simply down-weights high-frequency directions so the expectation stays finite. Specifically, choosing an orthonormal basis $(e_i)$ of $\mathcal{H}$ and nonnegative weights $(q_i)$ with $\sum_i q_i < \infty$, we can define $W_t^Q$ as the *convergent* series
$$W_t^Q := \sum_{i=1}^{\infty} \sqrt{q_i} W_t^i e_i,$$

Table 5: Summary statistics (mean $\pm$ std across 5 seeds) for all models under the identity (I) and optimal-transport (OT) couplings.

| Model | 1-Wasserstein distance | 2-Wasserstein Distance |
|-------|------------------------|------------------------|
| I-CFM | $11.84_{\pm 0.14}$ | $8.41_{\pm 0.10}$ |
| I-TFM | $\mathbf{8.99}_{\pm \mathbf{0.06}}$ | $\mathbf{6.42}_{\pm \mathbf{0.04}}$ |
| I-TAN | $11.24_{\pm 0.09}$ | $7.97_{\pm 0.06}$ |
| OT-CFM | $11.77_{\pm 0.08}$ | $8.36_{\pm 0.06}$ |
| OT-TFM | $\mathbf{8.97}_{\pm \mathbf{0.05}}$ | $\mathbf{6.41}_{\pm \mathbf{0.04}}$ |
| OT-TAN | $11.22_{\pm 0.09}$ | $7.95_{\pm 0.06}$ |

where $W_t^i$ are independent 1-dimensional Brownian motions. With such a choice the reference process

$$\mathrm{d}X_t = [H_t(\boldsymbol{L}_k)X_t + \alpha_t]\,\mathrm{d}t + \sigma\,\mathrm{d}W_t^Q$$

is well-posed, and the bridges remain Gaussian. All expressions for conditional controls, conditional paths, and transport costs from §4 carry over essentially verbatim. The zero-noise limit yields TFM on infinite simplicial complexes.

**Functions and differential forms on compact Riemannian manifolds**  The compact Riemannian manifold setting mirrors the simplicial one and can be treated exactly the same as the infinite simplicial complex setting under proper identifications. On a compact Riemannian manifold $\mathcal{M}$, we replace the space of $k$-simplex signals $\mathbb{R}^{n_k}$ with the space of $k$-forms $L^2\Lambda^k(\mathcal{M})$ and the Hodge Laplacian $\boldsymbol{L}_k$ with the Laplace–de Rham operator $\Delta^k$. Because $L^2\Lambda^k(\mathcal{M})$ has a countable basis $(e_i)$, we can again drive the reference SDE with a $Q$-Brownian motion, this time on $L^2\Lambda^k(M)$. The eigenfunctions of $\Delta^k$ have an analogous interpretation as wave-like signals for non-negative eigenvalues and signals circulating around "$k$-dimensional holes"; hence, the motivation for the topological reference process remains the same. Thus, under the identification $(\mathbb{R}^{n_k}, \boldsymbol{L}_k) \leftrightarrow (L^2\Lambda^k(\mathcal{M}), \Delta^k)$, the construction of TFM for signals on $\mathcal{M}$ can proceed exactly as for infinite simplicial complexes. An extension to functions on non-compact manifolds may be facilitated by taking functional flow matching (Kerrigan et al., 2024), or functional rectified flow (Zhang & Scott, 2026) as a starting point, possibly aided by the literature on the probability-flow ODE in function spaces (Na et al., 2025).

## D    FURTHER MOTIVATION OF THE TOPOLOGICAL REFERENCE PROCESS

This section expands on our motivation for the reference process used in TFM by comparison to alternative flow matching formulations based on a $Q$-Brownian motion that depends on the topology of the signal domain $K$. We provide a quantitative and qualitative comparison of these methods on a synthetic experiment, where $K$ is a triangulated torus.

**Alternative Topology-Aware Reference Processes.**  Topology awareness in flow matching could also be introduced by modifying the Brownian component of the reference SDE prior to taking the zero-noise limit. A natural candidate is a $Q$-Brownian motion,

$$dX_t = Q^{-1/2}dW_t,$$

where $Q$ depends on the Laplacian (e.g., $Q = L$). To investigate this idea, we analyze a more general reference process,

$$\mathrm{d}X_t = \sigma\, Q_t^{-1/2}\,\mathrm{d}W_t, \tag{29}$$

where $Q_t$ may vary in time and $\sigma \in \mathbb{R}_+$. Conditioned on endpoints $X_0 = x_0$ and $X_1 = x_1$, the process follows the bridge SDE

$$\mathrm{d}X_t = Q_t\Sigma_{t,1}^{-1}(x_1 - X_t)\,dt + \sigma Q_t^{-1/2}\,\mathrm{d}W_t, \qquad \Sigma_{s,t} = \int_s^t Q_r\,dr. \tag{30}$$

Taking the zero-noise limit $\sigma \to 0$, we obtain the deterministic bridge

$$X_t = x_1 + \Sigma_{t,1}\Sigma_{0,1}^{-1}(x_1 - x_0), \qquad \mathrm{d}X_t = Q_t\,\Sigma_{0,1}^{-1}(x_1 - x_0).$$

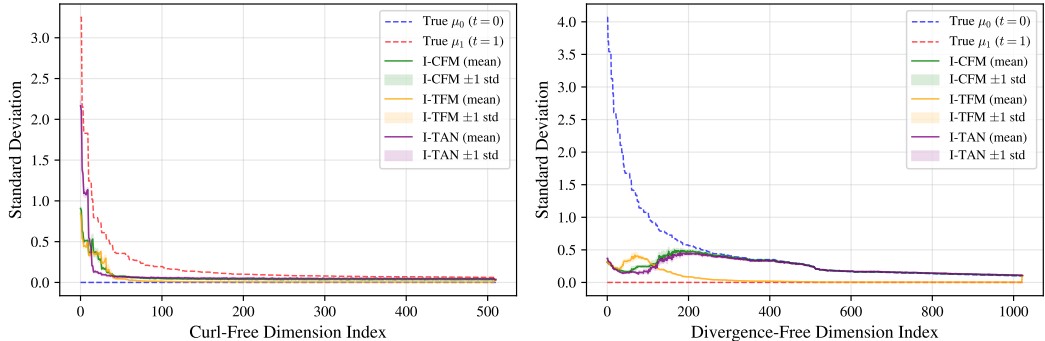

Figure 8: Standard deviation of the predicted distributions by the I-CFM, I-TFM, and I-TAN models in spectral coordinates. Left plot shows the results for coordinates corresponding to curl-free eigenvectors. Right plot shows the results for coordinates corresponding to divergence-free eigenvectors.

**Consequences for Flow Matching.** If $Q_t$ is time-homogeneous ($Q_t = Q_0$), then

$$Q_t \Sigma_{0,1}^{-1} = Q_0 \Big( \int_0^1 Q_0 \, \mathrm{d}s \Big)^{-1} = I,$$

so the resulting vector field is exactly $x_1 - x_0$, recovering *conditional flow matching* (CFM). Thus, constant-$Q$ topology-aware noising does *not* produce a different FM method.

When $Q_t$ is time-dependent, the zero-noise limit yields a genuinely different bridge. However, choosing a meaningful time-inhomogeneous $Q_t$ is nontrivial, in contrast to the physically motivated process

$$\mathrm{d}X_t = -cLX_t \, \mathrm{d}t + \sigma \, \mathrm{d}W_t,$$

which corresponds to heat diffusion perturbed by Brownian noise.

**A Time-Inhomogeneous Alternative.** For comparison, we consider

$$Q_t = \exp(-\kappa L t).$$

In the Laplacian eigenbasis this yields the conditional drift

$$(u_t(y)^{y_0, y_1})^i = \begin{cases} \dfrac{\kappa \lambda^i \exp(-\kappa \lambda^i t)}{1 - \exp(-\kappa \lambda^i)} (y_1^i - y_0^i), & \lambda^i > 0, \\ y_1^i - y_0^i, & \lambda^i = 0. \end{cases}$$

This expression resembles the TFM vector field but lacks the topology-dependent rescaling of $y_0^i$ via $\exp(-\kappa \lambda^i)$ that TFM introduces. For generation tasks from noise ($x_0$) to data ($x_1$), this mechanism effectively denoises low-frequency components before high-frequency ones.

**Synthetic Comparison on a Triangulated Torus.** To further motivate our approach and compare against this topology-aware noising (TAN) alternative, we conducted a controlled synthetic experiment on a triangulated torus. The task is to match distributions over edge signals: the initial distribution is divergence-free (no sources or sinks), and the target distribution is curl-free (no "swirls"). Specifically, the initial distribution is a Hodge-compositional Matérn Gaussian process (Yang et al., 2024) with smoothness parameter $\nu = 2.5$. We make this choice, instead of choosing the heat Gaussian process, to prevent making the task by-design easy for TFM and TAN.

We find that TFM outperforms both CFM and TAN in terms of the Wasserstein distance, as reported in Table 5. Furthermore, we can consider the standard deviation in the spectral coordinates of the final distribution modelled by each method. Specifically, by dividing the plots into the spectral dimensions corresponding to the curl-free and divergence-free components, we can understand how well these FM variants interpolate these qualitative differences of the boundary distributions. As shown in Figures 8 and 9, TFM dampens the curl-free features of the initial samples, introducing divergence-free flow in the final samples. In particular, TAN and CFM struggle to interpolate between the high-frequency components of the boundary distributions. These differences are qualitatively visible in sample visualizations in Figure 10.

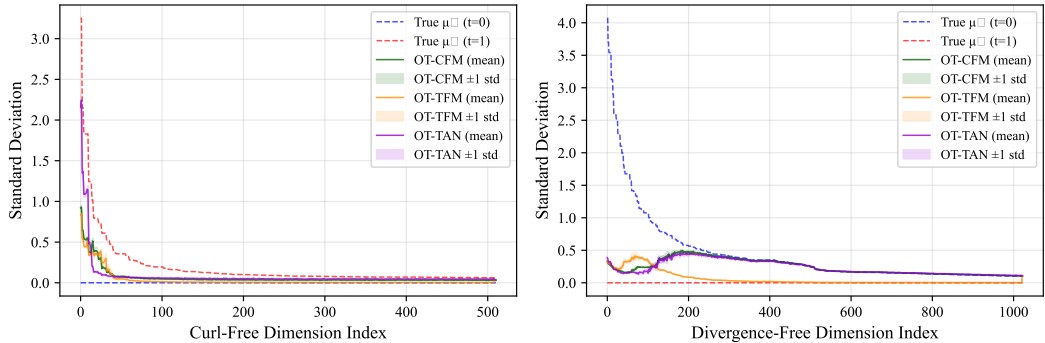

Figure 9: Standard deviation of the predicted distributions by the OT-CFM, OT-TFM, and OT-TAN models in spectral coordinates. Left plot shows the results for coordinates corresponding to curl-free eigenvectors. Right plot shows the results for coordinates corresponding to divergence-free eigenvectors.

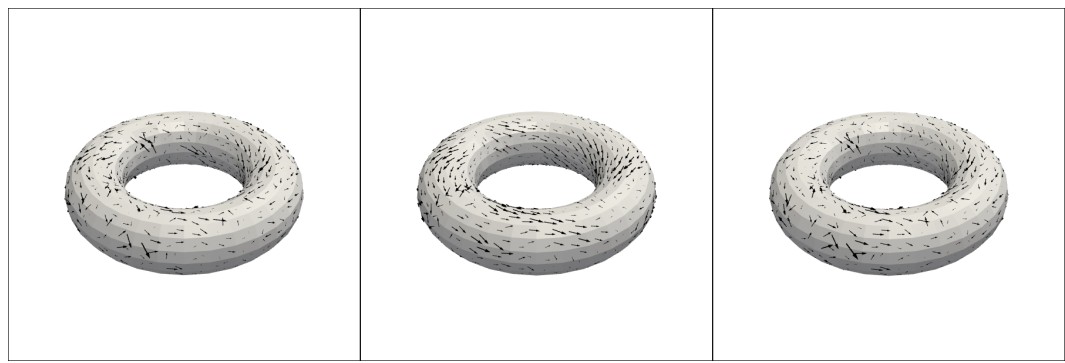

(a) Samples from the CFM (left), TFM (middle), and TAN (right) models.

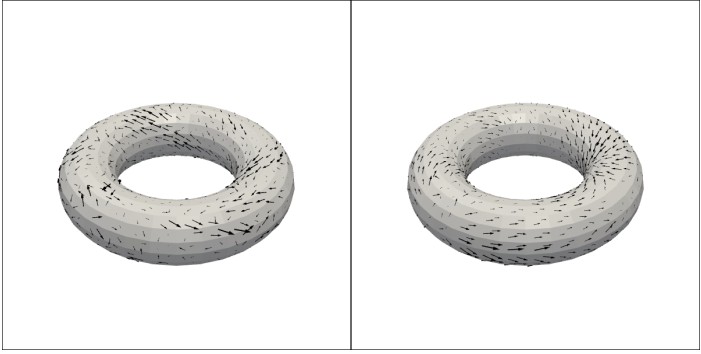

(b) Samples from the true initial (left) and final (right) distributions.

Figure 10: True and predicted samples in the synthetic matching experiment on the triangulated torus.

# E   ADDITIONAL EXPERIMENTAL DETAILS

## E.1   TOPOLOGICAL SCHRÖDINGER BRIDGE MATCHING EXPERIMENT SUITE

For the earthquake, traffic flow, brain fMRI, single-cell differentiation, and ocean currents experiments we replicate the experimental setup described extensively in Yang (2025). We summarize key aspects here, and provide additional details unspecified by Yang (2025) and ones that pertain to TFM specifically.

**Topological drift.** In all experiments we use the heat equation topological drift $-\kappa \boldsymbol{L}_k$ with $\kappa = 2.0$, where the value of $\kappa$ was set after initial testing on a synthetic dataset, though values from $0.5$ to $4.0$ performed similarly well. Depending on the size of the graph, we approximated the eigendecomposition of $\boldsymbol{L}_k$ with $m$ eigenpairs with the lowest eigenvalues: for earthquakes we take the full spectrum, for traffic flow the full spectrum, for brain the full spectrum, for single-cell differentiation $m = 256$, and for ocean currents $m = 500$.

**Training.** To learn the conditional control we trained a residual neural network, as well as a graph neural network and a simplicial neural network, depending on the signal domain, implemented by Yang (2025). Each training run consisted of up to 100 epochs with 25,600 samples each, stopping early after 10 epochs of improvement no greater than 1% in terms of 1-Wasserstein distance on a withheld validation set. For generation experiments and ocean current experiments, where at least one distribution is analytic, we approximate the optimal transport plan with batch-wise optimal transport (Tong et al., 2024).

**Evaluation.** In generation tasks, evaluation was done by sampling 512 points $X_0$ from the initial distribution $\mu_0$, obtaining predicted samples $\hat{X}_1$ by simulating the flow ODE started at $X_0$ and computing the 1- and 2-Wasserstein distances between their distribution and the test set. Averaging the result over 16 independent samples to obtain a final metric. For the ocean experiment we compute the metrics in the same way, except we also resample 512 points from the target distribution 16 times. For the single-cell and brain datasets we simply compute the Wasserstein distances exactly between the empirical distribution of the withheld target point and the initial points transported along the flow ODE.

### E.2 IMAGE GENERATION ON CIFAR-10

For image generation on the CIFAR-10 dataset we used the experimental setup of Tong et al. (2024). Because it is designed for CFM, we simply replaced its corresponding components according to Table 1. After initial testing of $\kappa \in \{0.001, 0.01, 0.1, 1.0\}$, we chose the best-performing TFM variant with $\kappa = 0.01$. We perform full eigendecomposition of the Laplacian, which can be done efficiently due to the product structure of the grid. All other setup pertaining to training and testing is done according to Tong et al. (2024). In particular, we use a UNet (Ronneberger et al., 2015) for learning of the conditional control and compute the FID scores using `clean-fid` (Parmar et al., 2022).

### E.3 ADDITIONAL RESULTS

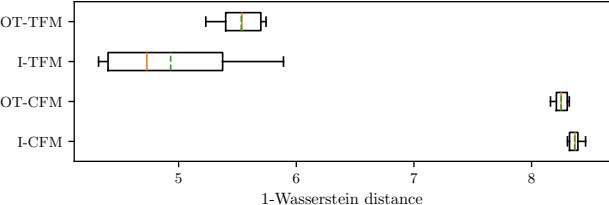

Figure 11: Test performance on the earthquake magnitude generation experiment, in terms of 1-Wasserstein distance measured over 10 independent runs. Orange bars show median value, green dashed bars show the mean, boxes show interquartile range, and outliers are shown as circles.

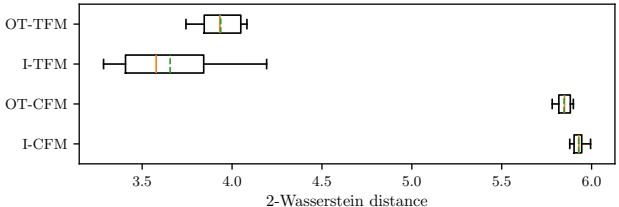

Figure 12: Test performance on the earthquake magnitude generation experiment, in terms of 2-Wasserstein distance measured over 10 independent runs. Orange bars show median value, green dashed bars show the mean, boxes show interquartile range, and outliers are shown as circles.

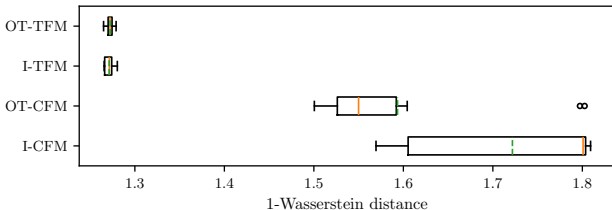

Figure 13: Test performance of topological and Euclidean flow matching models on the traffic flow generation experiment, in terms of 1-Wasserstein distance measured over 10 independent runs. Orange bars show median value, green dashed bars show the mean, boxes show interquartile range, and outliers are shown as circles.

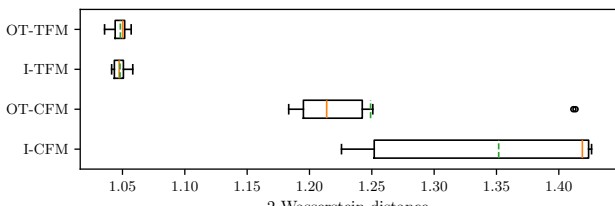

Figure 14: Test performance of topological and Euclidean flow matching models on the traffic flow generation experiment, in terms of 2-Wasserstein distance measured over 10 independent runs. Orange bars show median value, green dashed bars show the mean, boxes show interquartile range, and outliers are shown as circles.

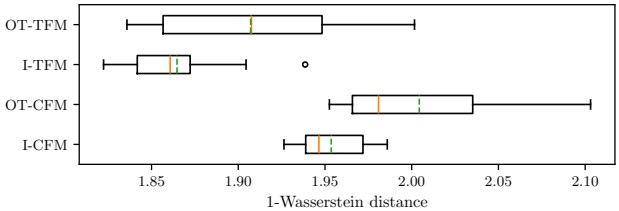

Figure 15: Test performance on the ocean current matching experiment, in terms of 1-Wasserstein distance measured over 10 independent runs. Orange bars show median value, green dashed bars show the mean, boxes show interquartile range, and outliers are shown as circles.

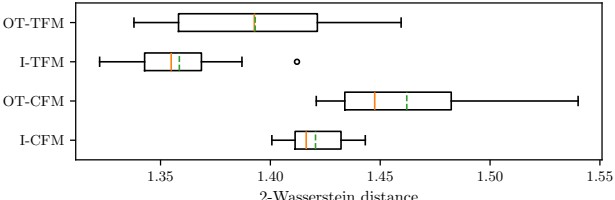

Figure 16: Test performance on the ocean current matching experiment, in terms of 2-Wasserstein distance measured over 10 independent runs. Orange bars show median value, green dashed bars show the mean, boxes show interquartile range, and outliers are shown as circles.

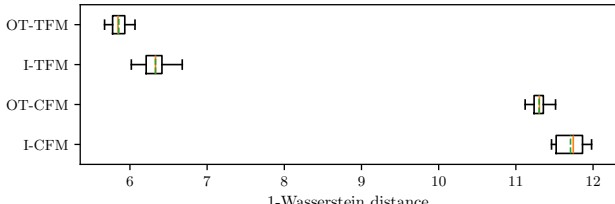

Figure 17: Test performance on the brain fMRI matching experiment, in terms of 1-Wasserstein distance measured over 10 independent runs. Orange bars show median value, green dashed bars show the mean, boxes show interquartile range, and outliers are shown as circles.

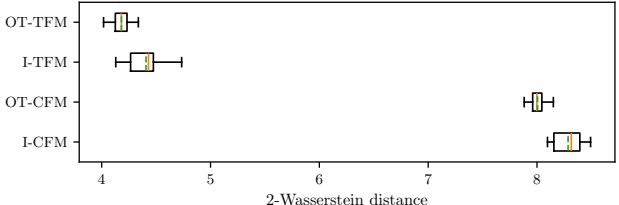

Figure 18: Test performance on the brain fMRI matching experiment, in terms of 2-Wasserstein distance measured over 10 independent runs. Orange bars show median value, green dashed bars show the mean, boxes show interquartile range, and outliers are shown as circles.

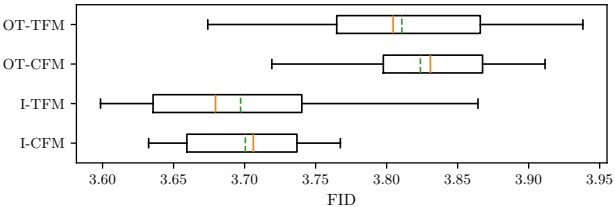

Figure 19: FID on the CIFAR-10 image generation experiment, computed over 10 independent runs. Orange bars show median value, green dashed bars show the mean, boxes show interquartile range, and outliers are shown as circles.

Table 6: Mean 1-Wasserstein distance, $\pm 1$ standard deviation, of CFM in spectral and standard coordinates on real-world datasets from (Yang, 2025).

| Method (coordinates) | Earthquakes | Traffic | Brain | Single Cell | Ocean Currents |
|---|---|---|---|---|---|
| I-CFM (Spectral) | $8.37_{\pm 0.05}$ | $1.72_{\pm 0.01}$ | $11.71_{\pm 0.02}$ | $0.022_{\pm 0.001}$ | $1.95_{\pm 0.02}$ |
| I-CFM (Standard) | $8.29_{\pm 0.05}$ | $1.76_{\pm 0.01}$ | $11.86_{\pm 0.23}$ | $0.020_{\pm 0.001}$ | $1.93_{\pm 0.02}$ |
| OT-CFM (Spectral) | $8.25_{\pm 0.06}$ | $1.59_{\pm 0.01}$ | $11.30_{\pm 0.01}$ | $0.019_{\pm 0.001}$ | $2.00_{\pm 0.05}$ |
| OT-CFM (Standard) | $8.26_{\pm 0.07}$ | $1.47_{\pm 0.18}$ | $11.50_{\pm 0.05}$ | $0.019_{\pm 0.001}$ | $1.98_{\pm 0.02}$ |

## F  SUPPLEMENTARY EXPERIMENTS AND ABLATIONS

### F.1  EFFECT OF COORDINATE FRAME ON CFM PERFORMANCE

One way to attempt an incorporation of topological information into the neural network $u_t^\theta$, is to predict the vector field in spectral coordinates $Y = \boldsymbol{U}_k^\top X$. Algebraically, the CFM conditional vector field in spectral coordinates takes the same form as in standard coordinates

$$u_t^{x_0, x_1}(X_t) = x_1 - x_0 = \boldsymbol{U}_k y_1 - \boldsymbol{U}_k y_0 = \boldsymbol{U}_k (y_1 - y_0) = \boldsymbol{U}_k u_t^{y_0, y_1}(Y_t).$$

Our empirical results reported in Table 6 suggest that there is no meaningful difference in performance between performing CFM in standard coordinates compared with spectral coordinates on the datasets from Yang (2025).

### F.2  EFFECT OF THE INITIAL DISTRIBUTION ON PERFORMANCE

Using the heat Gaussian processes $\exp(-\kappa \boldsymbol{L}_k)$ can boost performance of CFM and TFM in the generation experiments with earthquake magnitudes and traffic data. The performance gain is significant for CFM and relatively small for TFM. We report the results in Table 7.

Table 7: 1-Wasserstein distance for TFM and CFM compared across the normal and heat Gaussian process (GP) initial distribution on the traffic flows and earthquake magnitudes experiments.

| Dataset | Initial Distribution | I-TFM | OT-TFM | I-CFM | OT-CFM |
|---|---|---|---|---|---|
| Traffic | Normal | $1.30_{\pm 0.01}$ | $1.28_{\pm 0.01}$ | $1.72_{\pm 0.01}$ | $1.59_{\pm 0.01}$ |
| | Heat GP | $1.27_{\pm 0.01}$ | $1.27_{\pm 0.00}$ | $1.47_{\pm 0.01}$ | $1.45_{\pm 0.01}$ |
| Earthquakes | Normal | $5.35_{\pm 0.07}$ | $5.49_{\pm 0.10}$ | $8.37_{\pm 0.05}$ | $8.25_{\pm 0.06}$ |
| | Heat GP | $4.93_{\pm 0.06}$ | $5.53_{\pm 0.02}$ | $7.02_{\pm 0.07}$ | $7.39_{\pm 0.17}$ |

### F.3  EFFECT OF $\kappa$ ON PERFORMANCE

The parameter $\kappa$ in the topological drift $-\kappa \boldsymbol{L}_k X_t \, \mathrm{d}t$ may be further tuned to improve performance of TFM. This is shown for a range of $\kappa$ values across the earthquake magnitude, traffic flow, brain fMRI, single-cell differentiation, and ocean current experiments in Figures 20 to 24. Except for the single-cell experiment, it appears that the 1- and 2-Wasserstein distance is approximately convex in $\kappa$.

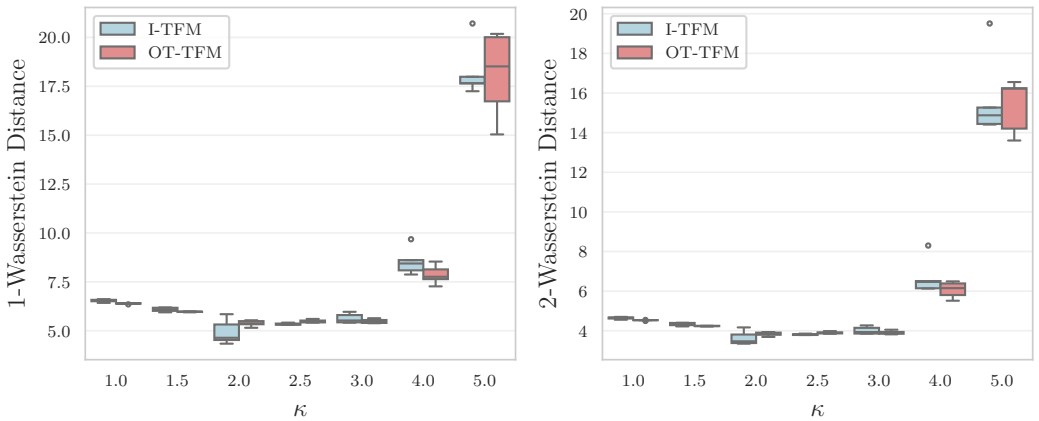

Figure 20: Test performance of I-TFM and OT-TFM across a range of $\kappa$ choices on the earthquake magnitude generation experiment over 5 independent runs. Bars show median value, boxes show interquartile range, and outliers are shown as circles.

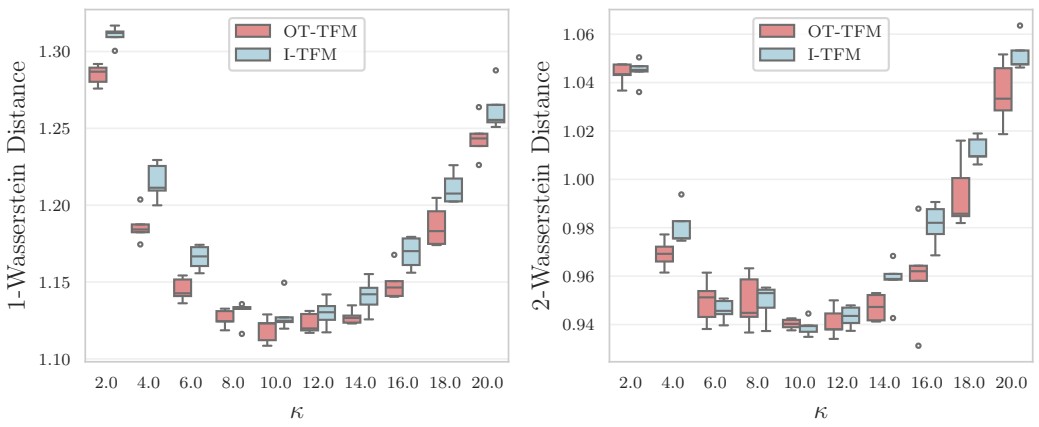

Figure 21: Test performance of I-TFM and OT-TFM across a range of $\kappa$ choices on the traffic generation experiment over 5 independent runs. Bars show median value, boxes show interquartile range, and outliers are shown as circles.

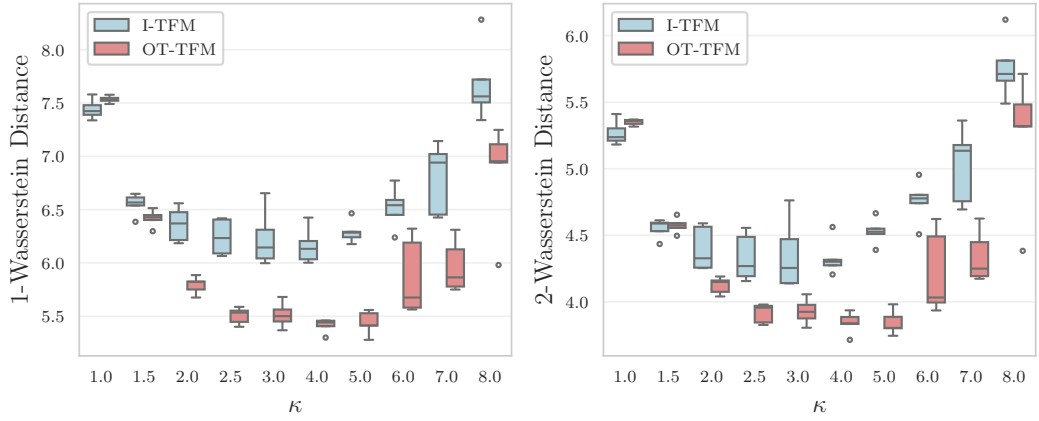

Figure 22: Test performance of I-TFM and OT-TFM across a range of $\kappa$ choices on the brain fMRI matching experiment over 5 independent runs. Bars show median value, boxes show interquartile range, and outliers are shown as circles.

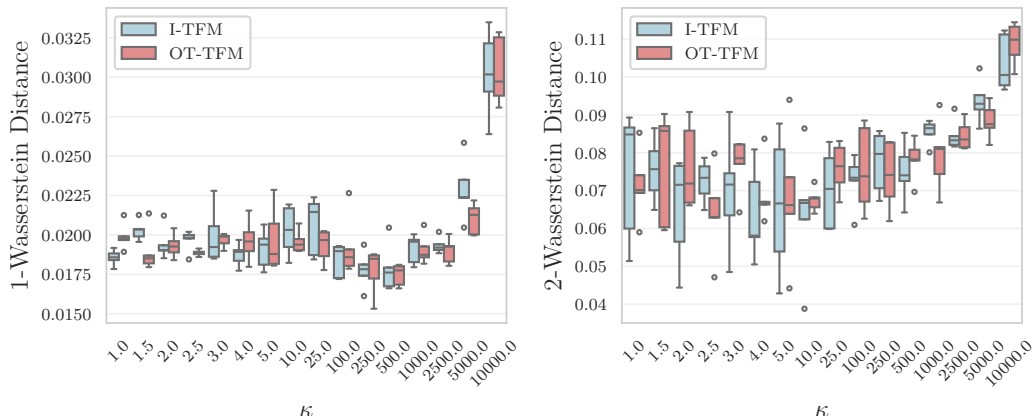

Figure 23: Test performance of I-TFM and OT-TFM across a range of $\kappa$ choices on the single cell differentiation matching experiment over 5 independent runs. Bars show median value, boxes show interquartile range, and outliers are shown as circles.

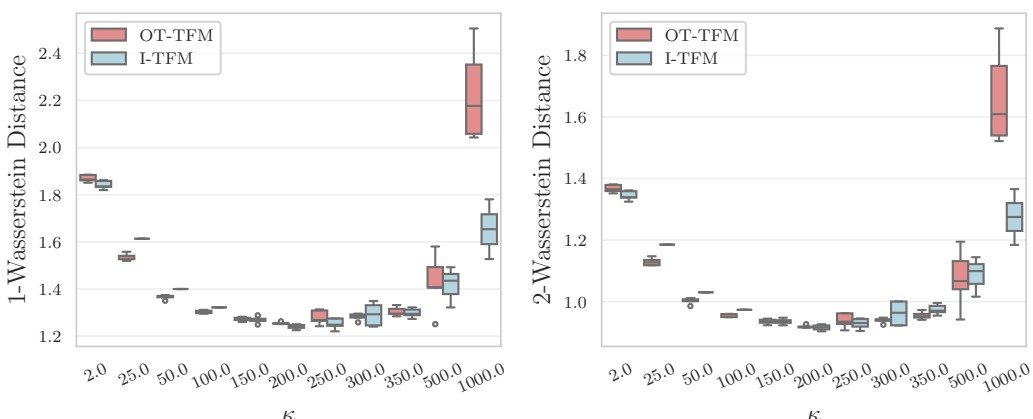

Figure 24: Test performance of I-TFM and OT-TFM across a range of $\kappa$ choices on the ocean current matching experiment over 5 independent runs. Bars show median value, boxes show interquartile range, and outliers are shown as circles.

