# OpenReview forum: "Topological Flow Matching"
_ICLR.cc/2026/Conference — ICLR 2026 Poster_

### Official Review · Reviewer_g1EL · 2025-10-25

**Soundness:** 3
**Presentation:** 1
**Contribution:** 1
**Rating:** 4
**Confidence:** 4

**Summary:**

The authors then derived topological flow matching based on the following two ideas or framework:
1. the unifying framework between the popular flow matching and Schrodinger bridge matching methods in terms of optimal transport (OT).
2. a piece of recent work generalizing the classical schrodinger bridge matching to the topological setting ---- where a brownian motion reference process is generalized by a topological diffusion driven reference process.

**Strengths:**

This paper studied the flow matching method in the topological setting and the experiments are diverse.

**Weaknesses:**

1. In my opinion, both methods, flow matching and bridge matching, have their pros and cons, where the former is more straightforward and efficient in terms of training, whereas the latter is more detailed in terms of modeling the intermediate distributions. Intermediate states modeling and learning are quite important in many engineering and science problems, e.g., single-cell dynamics, or the other trajectory related problems in this paper. For this reason, I think the paper can be improved by considering trajectory related tasks in topological settings, otherwise, it's hard to capture the advantage of flow matching besides improving the numbers.
2. I think the paper presentation can be improved by removing some previous results like flow matching and classical bridge matching and  elaborating the main topological related results, so you have more space to explain these results, instead of just layout tons of equations.

**Questions:**

1. I quickly skimmed the recent work by Yang 2025 on topological bridge matching, and noticed there is also a section about topological flow matching. I suggest authors to address the main contributions, difference or improvement of this work compared to the previous work.
2. In experiments, how do you optimize the diffusion coefficient $\kappa$? Could you elaborate experiment findings, in terms of, e.g., topological diffusion? Moreover, could we measure the training efficiency of flow matching compared to bridge matching? Without this, it's hard to see the advantage of flow matching besides improving the numbers. This is why I think the presentation can be improved by reducing some classical results in the main text so you have space to discuss your results and method.

---

> ### Author Response · Authors · 2025-11-27
> **Response to Reviewer g1EL Part 1**
>
> We thank the reviewer for their thoughtful feedback, recognizing the diversity of our experiments, and sharing thought-provoking questions. We address points made by the reviewer below.
>
>
> ## Motivating Flow Matching vs. Bridge Matching (Weakness 1)
>
> We thank the reviewer for sharing their views on the differences between flow matching and bridge matching. We agree that for problems where the ground truth dynamics are strictly stochastic (SDEs), SBM offers theoretical advantages in modeling intermediate dynamics. However, for many transport and generative tasks, the primary goal is an efficient mapping between distributions rather than recovering a specific stochastic path.
>
> We would like to politely push back on the statement that bridge matching is more detailed in modeling intermediate distributions. We don’t see how this can be the case due to the theoretical equivalence in modeling distributions of deterministic and stochastic dynamics. We also would like to politely note that our goal is to show that TFM improves the numbers, and that this even alone is enough to demonstrate the effectiveness and usefulness of our method.
>
> Nevertheless, we investigated the only trajectory-inference task that we know of in the topological context. Yang (2025) performs a trajectory inference task on the embryoid body dataset from Tong et al. (2020). We found that this task extends the single-cell experiment we already include, by mapping the predicted probability density at the final time into developmental stages using a binning procedure. We found that this evaluation assigns the lowest-probability nodes to the latest (day 24–27) time point, even though the true final distribution assigns its highest mass to these nodes; thus, models that place probability correctly may be penalized.
>
> When reproducing this experiment using the authors’ code, we further observed that a simple baseline—evolving the initial distribution under the topological heat equation—yields almost identical scores and predicted labels to those reported for TSBM. This suggests that, for this particular metric, the outcome is largely governed by the behaviour of the topological reference process rather than the learned bridge itself.
>
> Given this ambiguity, we chose not to include this task in our set of experiments. Nevertheless, we appreciate the reviewer's focus on a qualitative evaluation beyond the numbers, which we address below.
>
>
> ## Relevance of TFM beyond improving the numbers. (Question 2)
> We respectfully point out that improving the numbers is a key way of showing advantage over other methods, especially if the baseline performance is already good enough to be visually convincing. The Wasserstein distance is a true distance in the space of probability distributions; hence, a lower Wasserstein distance implies a meaningfully closer predicted distribution to the true data distribution. This allows us to unambiguously argue that TFM models the data distribution better than TSBM as well as flow matching methods that do not take topology into account.
>
> Nevertheless, we recognize the value of a multi-faceted evaluation. To support our findings beyond the improvement in metrics, we include a synthetic experiment, which allows us to measure how well TFM interpolates between two distributions of qualitatively different nature, as compared to CFM. In this controlled setting, we compare how TFM, CFM, and a topology-aware noise-based reference process transform a divergence-free distribution into a curl-free one on a triangulated torus. By analysing the spectra of the predicted distributions and visually inspecting the generated samples, we show that TFM provides a substantially smoother and more faithful interpolation, with markedly better handling of high-frequency components. This experiment demonstrates that the benefits of TFM extend beyond quantitative metrics, highlighting its ability to capture and respect the underlying geometric and topological structure of the data.

---

> ### Author Response · Authors · 2025-11-27
> **Response to Reviewer g1EL Part 2**
>
> ## Adding context to main results (Weakness 2)
> We appreciate the reviewer's perspective on optimizing the paper's focus.
>
> However, we respectfully believe that retaining the background on flow matching and the Schrodinger bridge problem is necessary to ensure that the paper remains self-contained and accessible to the broad ICLR audience, as there are many who may not be familiar with these topics in our community. This is especially important, since the key observation enabling our topology-aware framework is the intimate link between FM and the zero-noise limit of the SBP.
>
> Nevertheless, we fully agree that our paper would benefit from further discussion of the results related to the topological aspect of our work. On the theoretical end, we have expanded Section 3.1 with a clarification of why take the perspective of the zero-noise limit $\sigma \to 0$ of the SBP, rather than simply setting $\sigma=0$, showing a simple example of how the SBP generally fails to have a solution in that case. Moreover, in Section 3.2, we further motivate the choice of the topological drift, by explaining that a polynomial $H_t$ achieves tractability without sacrificing flexibility, as this choice can approximate any analytic function of $\mathbf{L}_k$. Additionally, in Section 4, we make it explicitly clear why the SBP is required to select a topology-aware control, and why a naive choice like $u_t^{x_0, x_1}(X_t) = (x_1 - x_0) - (H_t(L_t)X_t + \alpha)$ would fail to capture the topology by ignoring the Laplacian. Finally, in the same section we also expand the intuition and motivation behind our method by an equivalence to an optimal control problem, which shows that the control learned by TFM is the minimum-energy vector field needed to steer the initial distribution into the final distribution, if the undisturbed system evolves according to the heat equation.
>
> We believe these additions have greatly improved the motivation and presentation of our main contributions related to the topological aspect of TFM, for which we thank the reviewer.
>
> If the reviewer has any specific points they feel would benefit from further elaboration, we would be happy to provide this, especially given the additional content page allowed in the camera-ready version.
>
> ## Optimization of the diffusion coefficient $\kappa$ (Question 2)
> We thank the reviewer for highlighting the need for further elaboration on the coefficient $\kappa$.
> The diffusion coefficient is constant and set to $0.01$ in the CIFAR10 experiment and $2.0$ for the experiments from Yang (2025).
> Indeed, we did not focus on fine-tuning this coefficient, showcasing the advantage of our method on structured domains even with a simple heuristic choice like $2.0$.
> Nevertheless, the performance gain over CFM and TSBM can be further increased by fine-tuning the choice of $\kappa$ on each experiment, which we demonstrate in Appendix F.3, now referenced from the Experiments section.
> While parametric optimization of $\kappa$ may be viable during training of $u_t^\theta$, we keep it constant for a fair comparison with TSBM, reserving such an exploration of this direction for future work.
>
> We hope this addressed the reviewer's question and welcome any follow-ups.
>
>
> ## Elaboration of experimental findings in terms of topological diffusion (Question 2)
> We thank the reviewer for this suggestion.
>
> To the best of our knowledge, there are no topological diffusion models that would serve a comparison with our model. The published works on topology-aware diffusion models tackle generation of data with predetermined topological features (specifically Betti numbers) in images [1] and in 3d shapes [2]. In contrast, our work tackles the problem of generation of signals on topological spaces, and we have not found published work on dedicated diffusion models tackling this challenge.
>
> If there are works that we are unaware of, or the reviewer's intended meaning was different, we are happy to address this question further.
>
> [1] Gupta, S., Samaras, D., & Chen, C. (2025). TopoDiffusionNet: A Topology-aware Diffusion Model. International Conference on Learning Representations (ICLR).
>
> [2] Hu, J., Fei, B., Xu, B., Hou, F., Yang, W., Wang, S., Lei, N., Qian, C., & He, Y. (2024). Topology-Aware Latent Diffusion for 3D Shape Generation. arXiv:2401.17603.

---

> ### Author Response · Authors · 2025-11-27
> **Response to Reviewer g1EL Part 3**
>
> ## Measuring efficiency of TFM vs TSBM (Question 2)
> We thank the reviewer for this suggestion.
>
> We assume that the reviewer means the training efficiency of TFM vs TSBM. Such a comparison is made challenging, because we are unable to reproduce the results of Yang (2025) with the codebase supporting their work. Nevertheless, we tested the runtime of TFM vs TSBM until convergence across five independent runs, finding that TFM is approximately 8.8 times faster to train. The exact training time statistics for the brain fMRI experiment, running on an Apple M4 Max CPU, are reported below
> | Method | Mean (s) | Standard Deviation (s) |
> |--------|----------|---------|
> | OT-TFM | 32.4     | 3.2     |
> | TSBM   | 285.2    | 6.0     |
>
> This measurement is somewhat informal, considering that the TFM and TSBM implementations were not optimized for speed; however, our observations do align with prior work comparing simulation-free generative frameworks (like TFM) against simulation-based ones (like TSBM)---e.g. [1] reports at least ~12.7 times speed-up of CFM vs simulation-based continuous normalizing flows.
>
> [1] Tong, A., Fatras, K., Malkin, N., Huguet, G., Zhang, Y., Rector-Brooks, J., Wolf, G., & Bengio, Y. (2024). Improving and Generalizing Flow-Based Generative Models with Minibatch Optimal Transport. Transactions on Machine Learning Research. arXiv:2302.00482.
>
>
> ## Differentiation from Yang (2025) (Question 1)
> We thank the reviewer for the opportunity to clarify our contribution relative to Yang 2025.
>
> Beforehand, however, we would like to address the question of whether topological flow matching is discussed in Yang (2025). We are not sure which section of that work the reviewer might be referring to. Indeed, in correspondence with the work's author, we confirmed that such a topological flow matching framework is not directly discussed. Our best guess is that the reviewer may be referring to Corollary E.2, which gives the probability flow ODE for the solution to the topological SBP. The drift in this formula, however, is given in terms of 2 random variables determined by a coupled system of two heat equations, which makes it computationally intractable and inapplicable for a flow matching framework.
>
> Regarding the differences from TSBM, we emphasize that our method enjoys simulation-free training whereas TSBM is simulation-based. This yields a large empirical improvement in training speed and stability, as well as making training conceptually simpler and more scalable. We also highlight the qualitative difference between sample paths: TFM paths are deterministic, while TSBM paths are stochastic. In short, because TFM retains the key properties of FM (scalable, simulation-free training with deterministic samples paths), while TSBM retains those of SBM (simulation-based training with stochastic sample paths), the differences between them are equivalent to the differences between FM and SBM. Additionally, we suggest a topology-aware initial distribution for generation tasks, which can improve performance, as is now shown in Appendix F.2.
>
> We have conveyed the differences discussed above in a focussed discussion in Appendix B.1. We believe this addition helps clarify the contribution of our work, for which we thank the reviewer.
>
>
> ## Conclusion
> We thank the reviewer for the helpful suggestions in clarifying the contributions of our work. We believe these clarifications and additional experiments have improved the clarity of the benefits of our approach, particularly against recent works such as Yang et al. 2025. If the reviewer agrees we would be grateful if the reviewer considered updating their review to reflect these clarifications.

---

### Official Review · Reviewer_Dei1 · 2025-10-29

**Soundness:** 3
**Presentation:** 1
**Contribution:** 3
**Rating:** 2
**Confidence:** 3

**Summary:**

This work proposes topological flow matching model, which is derived by first solving a Schrodinger Bridge Problem with drift, and then taking the noise scale to 0. The proposed model is designed to model data with a graph structure. The math is carefully derived and experiments are conducted on several topological datasets with the topological Schrodinger Bridge Matching as the only baseline.

**Strengths:**

1. This work proposes a novel generative framework for data embedded in graphs.
2. The proposed model can be trained simulation free and samples can be generated by solving ODE.
3. This work reveals an interesting connection between Schrodinger Bridge and flow models, i.e., a flow model can be obtained by solving a SBP with diminishing diffusion noise.

**Weaknesses:**

First, I think the overall presentation is unsatisfying to the extent that I'm not able to appreciate some of the authors' major claims. Especially,

* The word "topology" is not precisely defined in the paper. I could only guess from the context that it is related to the graphical structure of the data, which does not seem to be a standard definition for me (For me, the standard definition of topology refers to a topological space in analysis).

* It is not clear to me where does the matrix $L_k$ come from and how the data $X_1$ is related to the matrix $L_k$. From the context, I can only infer that $L_k$ is determined by some domain experts or auxiliary knowledge and is independent from data $X_1$.

* As a consequence of 1 and 2, it is not clear to me on how the flow ODE ignores the topology while the proposed method retains it around line 247, which seems to be an important advantage of the proposed method.

On the other hand, I think the empirical validation is insufficient. While this work claims standard FM in Euclidean space ignores the rich graph structure on data, it does not empirically compare the proposed method against generative models in Euclidean space.

Additionally, the equation on line 220 seems to suggest that we can diagonalize the flow with a change of coordinates. If so, what is the advantage of the proposed method against  1. performing change of coordinates 2. then applying generative models in Euclidean space. 3. transforming the generated data back to the original coordinates when sampling.

For the ACs and the authors to better understand my position, I'm familiar with SDE, diffusion, and flow models and I have some exposure to Schrodinger Bridge literature. But I have no working knowledge on data with a graph structure.

I'm happy to increase the score if the authors could help me better appreciate their work.

**Questions:**

1. how should I interpret the SBP when $\sigma \to 0$? It seems like the reference process then becomes an ODE? What if you just solve the SBP with zero-noise? Would it still be a well-posed optimal transport problem? If so, will it lead to the same vector field on line 246?

2. What motivates the particular form in (11) and $H_t$ as a polynomial?

3. The form of (11), the change of coordinate interpretation (around line 220), and the infinite dimensional generalization (line 781-795) in the appendix altogether present significant resemblance in form with some recent works on diffusion/flow models on functional space, like [1, 2, 3, 5]. Do you think there is any potential connections? Maybe the functional models can be deduced from the SBP perspective or your proposed model can be considered as a special instance of functional model on non-Euclidean space?

4. Isn't line 134 also called rectified flow [4]?


References:

[1] Zhang, J., & Scott, C. (2025). Flow Straight and Fast in Hilbert Space: Functional Rectified Flow. arXiv. https://arxiv.org/abs/2509.10384

[2] Kerrigan, G., Migliorini, G., & Smyth, P. (2023). Functional Flow Matching. arXiv. https://arxiv.org/abs/2305.17209

[3] Na, K., Lee, J., Yun, S.-Y., & Lim, S. (2025). Probability-Flow ODE in Infinite-Dimensional Function Spaces. arXiv. https://arxiv.org/abs/2503.10219

[4]Liu, X., Gong, C., & Liu, Q. (2022). Flow Straight and Fast: Learning to Generate and Transfer Data with Rectified Flow. arXiv. https://arxiv.org/abs/2209.03003

[5]Franzese, G., Corallo, G., Rossi, S., Heinonen, M., Filippone, M., & Michiardi, P. (2023). Continuous-Time Functional Diffusion Processes. arXiv. https://arxiv.org/abs/2303.00800

---

> ### Author Response · Authors · 2025-11-27
> **Response to Reviewer Dei1 Part 1**
>
> We thank the reviewer for recognizing the novelty of our method, the advantages of simulation-free training, and the value of our perspective on flow matching via the SBP. We appreciate the reviewer's detailed feedback and apt questions, which we address below.
>
>
> ## Lack of precise definition of "topology" (Weakness 1)
> We thank the reviewer for highlighting the need for clarity in the meaning of "topology" in our work.
>
> Indeed, we left the meaning of the word "topology" at an intuitive level, instead of defining it in the analytic sense of a collection of open subsets closed under arbitrary unions and finite intersections. This is because graphs and simplicial complexes, although of central interest to topological deep learning, are not topological spaces in the sense of analysis---they are equipped with combinatorial, rather than topological, structure. Nonetheless, there is an unambiguous meaning to the "topological features" of such spaces in the analytic sense.  As is now described in Footnote 3, this is because any simplicial complex $K$ can be identified with a polyhedral subspace of a Euclidean space, called its geometric realisation, and any two geometric realisations of $K$ are homeomorphic---that is, topologically equivalent. Knowing that such an identification is valid allows us to meaningfully talk about the topological features of $K$ in the usual intuitive sense of properties preserved under continuous deformations like stretching or twisting, but not discontinuous ones like cutting or gluing. Thus, we can leave the tools _analytic_ topology, and instead study the topological features of $K$ through purely combinatorial tools of _algebraic_ topology such as the boundary matrices and Hodge Laplacians.
>
> In short, we use the word "topology" for a simplicial complex $K$ in the sense of algebraic topology which is made precise by the identification with a polyhedral subspace of $\mathbb{R}^d$. We feel that a rigorous mathematical exposition of this meaning would require mathematical detail that would make our work less accessible to the broad community---this is also why we refrain from defining the notions of cohomology groups precisely. Nevertheless, we agree with the reviewer's position and, aiming for maximum clarity without sacrificing accessibility, we now explain the intuitive meaning of "topological features" of simplicial complexes around line 115, providing a justification for the identification of the algebraic and analytic settings in Footnote 3.
>
> We sincerely appreciate the reviewer highlighting this matter, since we believe the above additions make the meaning of topology in our work significantly clearer and help the reader build useful intuitions. Of course, we happily welcome any further suggestions.
>
>
> ## Provenance of the Hodge Laplacian and its relation to data. (Weakness 2)
> We thank the reviewer for stressing the need for a clear explanation of where the Hodge Laplacian comes from and of its relation to data.
>
> The data $X_1$ is a collection of signals on the $k$-simplices of a given simplicial complex $K$. The Hodge Laplacian $\mathbf L_k \coloneqq \mathbf B_k^\top \mathbf B_{k} + \mathbf B_{k+1} \mathbf B_{k+1}^\top$ is determined by the structure of $K$. Specifically, it depends on how $k$-simplices connect to $(k-1)$-simplices (via $\mathbf B_{k}$) and how $(k+1)$-simplices connect to $k$-simplices via $\mathbf B_{k+1}$. Therefore, $\mathbf B_\bullet$ does not depend on the data $X_1$. Hence, by its definition, the Hodge Laplacian is also independent of $X_1$. In other words, if the simplicial complex is known, $\mathbf L_k$ is fully determined and requires no further domain or auxiliary knowledge. To make this clearer in the text, we moved the definition of the Hodge Laplacian from an inline to a standalone equation, and added an explicit statement that $\mathbf L_k$ is fully determined by the structure of $K$.
>
> That said, the structure of the simplicial complex is indeed often determined by experts, especially in real world application. In modeling brain fMRI signals, the nodes of the graph correspond to functional regions of the brain, which are determined by experts, thought the connectivity is determined more automatically, being inversely proportional to the square of the distance between nodes. On the other hand, in the traffic flow experiment, the structure of the underlying graph is determined by the road network, while the 2-simplices are added programmatically by filling in all 3-cliques. In order to build intuition and put other theoretical setup in a practical context, we added a discussion of the provenance of $K$ in the "Simplicial Complexes" paragraph of Section 2.1.
>
> We believe these additions have significantly improved the clarity of presentation and the intuition behind our work, for which we thank the reviewer.

---

> ### Author Response · Authors · 2025-11-27
> **Response to Reviewer Dei1 Part 2**
>
> ## Why does the flow ODE in line 247 ignore the topology while TFM retains it? (Weakness 3)
> We thank the reviewer for bringing this question to our attention.
>
> We agree that the reason why the proposed method around line 247 ignores the topology could be spelled out more clearly. To this end, we expanded the corresponding part of Section 4, explaining this matter precisely. Specifically, given the conditional control of the form $u^{x_0, x_1}_t(X_t) = (x_1 - x_0) - (H_t(\mathbf L_k)X_t + \alpha_t)$, the Laplacian term in the drift of the bridge $X_t \mid X_0 = x_0, X_1 = x_1$ vanishes
> $$
> \dot{X}_t = (H_t(\mathbf L_k)X_t + \alpha_t) + u_t^{x_0, x_1}(X_t) = (H_t(\mathbf L_k)X_t + \alpha_t) +  (x_1 - x_0) - (H_t(\mathbf L_k)X_t + \alpha_t) = x_1 - x_0,
> $$
> resulting in a conditional vector field of CFM. The CFM lacks any information about the topology of $K$. To put this precisely, regardless of the connectivity of simplices in $K$, the conditional vector field is the same. This is in contrast to our method, where the conditional vector fields vary consistently with $K$'s topology. We believe this additional explanation aids the presentation and motivation of the paper, for which we thank the reviewer.
>
> ## Lack of empirical validation against generative models in Euclidean space. (Weakness 4)
> We thank the reviewer for stressing the importance of validation against Euclidean baselines. We hope that the additional explanations in our manuscript have made it clear in what sense the Laplacian captures the topology and hence that CFM ignores the topology, effectively operating in Euclidean space. Indeed, CFM is not affected by the topology, since regardless of the signal domain, CFM uses the same conditional vector fields $u_t^{x_1, x_0}(x) = x_1 - x_0$, while OT-CFM uses the same optimal transport cost $c(x_0, x_1) = ||x_1 - x_0||^2$. Moreover, CFM may be adequately described as Euclidean, since $x_1 - x_0$ is the velocity field of a the short path from $x_0$ to $x_1$ in Euclidean space and $||x_1 - x_0||^2$ is the squared distance in Euclidean space. Thus, all our experiments compare our topology-aware framework against the Euclidean I-CFM and OT-CFM. We hope the reviewer's concern is alleviated and welcome any further questions.

---

> ### Author Response · Authors · 2025-11-27
> **Response to Reviewer Dei1 Part 3**
>
> ## Since the flow can be diagonalized in spectral coordinates, what is the advantage of TFM against Euclidean FM in spectral coordinates? (Weakness 5)
>
> Thank you for this apt question.
>
> Before we consider Euclidean FM in spectral coordinates, we should clarify in what way TFM can be diagonalized.
> It is true that the drift on line 220 is diagonal in the spectral coordinates.
> However, the flow learned by TFM generally may not be diagonalizable in spectral coordinates.
> Explicitly, while the conditional vector field is diagonalized as
> $$
> \mathbf U_k^\top u_t^{x_0, x_1}(x) = \mathrm{diag}(u_t^{y_0^i, y_1^i}(y^i)),
> $$
> the marginal vector field is given by
> $$
> \left(\mathbf U_k^\top \mathbb E_{(X_0, X_1)\sim\pi(\cdot \mid X_t = x)}\left[u_t^{X_0, X_1}(x)\right]\right)^i = \mathbb E_{(Y_0, Y_1)\sim\pi(\cdot \mid Y_t = y)}\left[u_t^{Y_0^i, Y_1^i}(y^i)\right],
> $$
> and the distribution $\pi(\cdot \mid Y_t = y)$ may not factor over dimensions. This is why the prior flow diagonalizes, but the learned flow generally does not.
>
> Nevertheless, the question of using a Euclidean model (I-CFM or OT-CFM) in spectral coordinates is still valid. Indeed, if data on simplicial complexes happens to have a simpler---for example, sparser---form in spectral coordinates, then CFM in spectral coordinates could have an advantage over CFM in Euclidean coordinates. Nevertheless, unless the neural network $u^\theta$ is designed explicitly to capture this simpler form---for instance, by having a lower-dimensional input and output space---it is unclear whether such an advantage would be realised. We tested this question empirically on the datasets from Yang (2025), comparing the performance of CFM in Euclidean and spectral coordinates. The results, summarised in the table below, do not show any significant performance difference between the coordinate systems.
>
> | Method (coordinates)  | Earthquakes       | Traffic          | Brain            | Single Cell        | Ocean Currents     |
> |------------------------|-------------------|------------------|------------------|---------------------|---------------------|
> | I-CFM (Spectral)       | 8.37 ± 0.05       | 1.72 ± 0.01      | 11.71 ± 0.02     | 0.022 ± 0.001       | 1.95 ± 0.02         |
> | I-CFM (Standard)       | 8.29 ± 0.05       | 1.76 ± 0.01      | 11.86 ± 0.23     | 0.020 ± 0.001       | 1.93 ± 0.02         |
> | OT-CFM (Spectral)      | 8.25 ± 0.06       | 1.59 ± 0.01      | 11.30 ± 0.01     | 0.019 ± 0.001       | 2.00 ± 0.05         |
> | OT-CFM (Standard)      | 8.26 ± 0.07       | 1.47 ± 0.18      | 11.50 ± 0.05     | 0.019 ± 0.001       | 1.98 ± 0.02         |
>
> We report these results in Appendix F.1, which is now referenced from Section 5.
>
> Indeed, performing Euclidean flow matching in spectral coordinates does not have the topology-aware properties of TFM, where the reference process serves as a smoothing bias preserving components of functions aligned with the topological features of the simplicial complex. Moreover, while the OT cost of TFM is topology-aware, the OT cost of CFM in spectral coordinates is exactly equal to the OT of CFM in standard coordinates, since the Euclidean distance is invariant to a change of coordinates.
>
> Nevertheless, while TFM boasts the same advantages over CFM, regardless of the coordinate system, the question of the impact of coordinate system on performance on a given dataset is intriguing. We believe the experimental comparison of CFM in standard and spectral coordinates makes the contributions of our work clearer, for which we thank the reviewer.

---

> ### Author Response · Authors · 2025-11-27
> **Response to Reviewer Dei1 Part 4**
>
> ## Well-posedness and interpretation of SBP in the zero-noise limit. (Question 1)
> We thank the reviewer for the deep question.
>
> Indeed, in the case $\sigma=0$, the reference process becomes an ODE. However, an SBP with an ODE reference process is generally not well-posed, as its set of candidate solutions may be empty. To see why, consider the SBP with the trivial reference ODE $\partial_t X_t = 0$ and an initial distribution $\delta_{x_0}$; thus, the reference law is a point mass on the constant path at $x_0$. Because a candidate law must be absolutely continuous with respect to the reference law, it too must be a point mass on the constant path at $x_0$, so its marginal at $t=1$ is $\delta_{x_0}$. Hence, unless the final distribution in the SBP is $\delta_{x_0}$, the SBP does not have a solution. This matter is now explained in Section 3.1 of the manuscript.
>
> Nevertheless, while SBP with $\sigma=0$ is generally ill-posed, we can give a meaningful interpretation of its limit. Indeed, by the equivalence between the Monge-Kantorich and Benamou-Brenier formulations of OT, we find that the control $u_t$ of the SBP solution, solves the optimal control problem
> $$
> \min \int_0^1 \tfrac{1}{2} ||u_t(x)||^2 \, \mathbb{P}_t(\mathrm{d}x)\, \mathrm{d}t,
> \qquad \text{s.t.} \quad
> \partial_t \mathbb{P}_t + \nabla \cdot \big(\mathbb{P}_t ( H_t(\mathbf{L}_k) + \alpha_t + u_t )\big) = 0,
> \qquad \mathbb{P}_0 = \mu_0,\quad \mathbb{P}_1 = \mu_1.
> $$
> Thus, intuitively, $u_t$ is the minimum-energy corrective force needed to steer from $\mu_0$ to $\mu_1$ when the system, left undisturbed, evolves according to the drift of the reference process. We now describe this viewpoint in Section 3.1, in the context of CFM, and in Section 4 in the context of TFM.
>
> We believe the additions listed above greatly improve the presentation and motivation of the paper, building intuitive understanding of TFM, for which we thank the reviewer.
>
>
> ## Motivation behind polynomial form of the topological drift. (Question 2)
> We thank the reviewer for raising the issue of the motivation behind the drift form.
>
> The motivation, as in TSBM, is tractability (since a polynomial of a diagonalizable matrix is diagonalizable) and flexibility. Specifically, by the Cayley-Hamilton theorem, any analytic function of $\mathbf{L}_k$ can be approximated to arbitrary precision by a suitable choice of $H_t$. This is now explained in Section 3.2.
>
> We believe this addition improves the clarity of our work, and we thank the reviewer for motivating this addition.
>
>
> ## Relation of TFM to functional models. (Question 3)
> We thank the reviewer for pointing us in this intriguing direction.
>
> It appears that TFM is directly related to functional flow matching (FFM) [1] and functional rectified flow (RFM) [2] and somewhat indirectly to the functional probability flow ODE [3] and functional diffusion [5], though it is possible we misunderstood something about these works. However, if we understand correctly, all of these methods assume the setting of a separable Hilbert space, which implies the existence of a Q-Wiener process. Thus, barring some functional analytic considerations, FFM (and RFM) can be viewed as solving the zero-noise limit of the SBP with a Q-Wiener reference process. Consequently, as we suggest in the Appendix, TFM should be able to extend FFM and RFM by augmenting the drift with a topological drift, which we now reference in the manuscript. Thus, TFM on compact Riemannian manifolds and countably infinite simplicial complexes, can be seen as a topology-aware generalization of FFM and RFM for these spaces, and could be applied to any separable Hilbert space equipped with a Laplacian.
>
> We hope this answers the reviewer's question and welcome any follow-ups.
>
>
> ## Equivalence between CFM and rectified flow. (Question 4)
>
> We thank the reviewer for bringing this to our attention. The equivalence between CFM and rectified flow is now noted in line 123.
>
>
> ## Conclusion
>
> We thank the reviewer for the careful reading and for raising points that prompted substantial clarification. The additions on the meaning of topology, provenance of the Laplacian, the role of the SBP, the diagonalization question, and the relation to functional models considerably strengthen the paper’s accessibility and conceptual framing. We hope these revisions help address the reviewer’s concerns, and we would appreciate any reconsideration of the score in light of the improved presentation.

---

### Official Review · Reviewer_8d47 · 2025-10-29

**Soundness:** 4
**Presentation:** 1
**Contribution:** 4
**Rating:** 4
**Confidence:** 3

**Summary:**

This paper extends Flow matching to the  general topological structure on which the Hodge Laplacian is well defined. The paper presents the flow matching as the variance0  limit of the Schrodinger bridge, and extends it to the Simplicial complex in a computable form by changing the reference process that acts as the regularizing anchor from which the solution flow can not diverge too greatly. The reference process is changed from the usual Wiener process to a process with Laplacian drift.
Most importantly , the paper's proposed loss objective maintains that the solution of their topological analogue of the Schrodinger bridge problem can be computed with the coupling that can be optimized with the loss of the closed form.
The method is shown competitive on various datasets.

**Strengths:**

This paper not only tackles an important problem of extending FM to general geometry (not necessarily Riemannian), it showcases a beautiful analogue between the propositions and the classical results. The way that the paper is narratin/organizing the classical works are very instructive and that on its own is a contribution to the community.
The method itself seems impactful when polished with appropriate engineering techniques, showing much potential for future applications.
The idea of changing the reference process to establish the unifying method with consistency across different topology in a scalable way, is also an approach that I as a researcher as waiting for a long time as well, because the  "wiener reference process" has long been adopted as a convention that does not always match the data-domain of interest.

**Weaknesses:**

I believe that the paper suffers much from the presentation.


1. From place to places, the conditions of the theoretical results are "embedded" in the paragraphs, for instance, in the first subsection of 3.1 (1. Conditional vector field ....  The diffusion briged in Eq 7 simplifies to)... shall, at least in my understanding, read as "when $b=0$ and $\sigma_t=\sigma_0$, the diffusion bridge SDE in equation 7 simplifies to".  Indeed, this is stated at the head of the 3.1, but it is separated by the section header and it took a while for me to make the connection.  I got lost several places through out, I would very much appreciate if the preambles of the theoretical statements are more complete and self contained inside each "thm environment".

2. Also, the reason of using the reference process of the proposed form seem to lack strong enough motivation.  I buy that this choice leads to very clean results and introduces a topological heat equation flavor, but why this choice, more precisely?   Using a refernece process "with drift" seems like a wild card that I have never seen out of finance. The effort  of the justification seems to be made in 3.2 but it did not click with me----why mix Wiener with (11) instead of completely replacing it with, say, a  corresponding stochastic analogue of heat equation in the topological space (maybe a one driven with an ensemble of random walk)?  Is additive mix with standard wiener process a logical choice?  What extreme case is this choice supposed to resolve?
 Because the paper is presenting a method that I believe, is intended to merit the applied community of wide range, I believe that the audience deserves further elaboration.   Correspondence table of comparison against standard Schrodinger may also be helpful.


3. In a related note with 2, a demonstrative extreme case experiments produced by synthetic data might have been helpful.


4.  Figure 2 did not appear too demonstrative to me, what advantageous aspect of the method is this figure supposed to convey?

**Questions:**

1. My greatest question is posed in the part 2 of the weaknesses.

2. Would you please mention more about the hyperparameter kappa?  This shall be quite important as it desribes the speed of the diffusion. The choice of this parameter may also relate to question1.

**Details Of Ethics Concerns:**

Nothing in particular.

---

> ### Author Response · Authors · 2025-11-27
> **Response to Reviewer 8d47 Part 1**
>
> We thank the reviewer for the detailed feedback, recognizing the merit in the synthesis of prior works in the buildup to TFM, the potential for impact in application of our method, and the value of the core idea of the topologic-aware reference process.
> We appreciate the reviewer's focus on precision in theoretical statements and the motivation behind the reference process, as well as the question about the hyperparameter $\kappa$.
>
>
> ## Clarity of the theoretical presentation (Weakness 1)
> We thank the reviewer for bringing this issue to light.
>
> To make the presentation of theoretical results more clear and self-contained, we now include the assumptions directly in their statements. Specifically, we expanded the preambles of Propositions 2., 3., and 4. to specify that $X$ is the topological reference process—whose definition has been made more pronounced in Section 3.2—and made explicit the conditions of $b = 0$, $\sigma \in \mathbb{R}_+$, and $\sigma \to 0$ at appropriate places in points 1., 2., 3. in Section 3.1.
>
> We thank the reviewer for the suggestion and believe this has greatly improved the flow and clarity of the theoretical presentation.

---

> ### Author Response · Authors · 2025-11-27
> **Response to Reviewer 8d47 Part 2**
>
> ## Further Motivation of the Reference Process (Weakness 2 and 3, and Question 1)
> We thank the reviewer for highlighting the importance of motivation behind the reference process.
>
> The motivation for our choice of reference process came from the strong performance gains it achieved for TSBM over standard Schrodinger bridge matching.
> Specifically, we noticed that the correspondence between FM and SBP allows us to employ the same topological reference process as TSBM to derive a topology-aware extension of FM, which could achieve analogous performance gains.
> Furthermore, since without noise the prior reduces to heat diffusion, we further motivated this process theoretically as a topology-aware smoothing bias.
>
> Nevertheless, as aptly suggested, this may not be the only way to make FM topology-aware. Indeed, we may also modify the Brownian motion part of the SDE before taking the zero-noise limit.
> A natural choice could be a $Q$-Brownian $Q^{-1/2} d W_t$, where $Q$ depends on the Laplacian—most notably, $Q=L$.
> However, while this is a natural stochastic process to consider, the solution of the SBP with this reference process is the same, in the zero-noise limit, as for standard Brownian motion.
> To see why, we can consider a more general reference process with a possibly time-dependent $Q$
> $$
> dX_t = \sigma Q_t^{-1/2} dW_t.
> $$
> Conditional on $X_0 = x_0, X_1 = x_1$, this process follows the SDE
> $$
> dX_t = Q_t \Sigma_{t, 1}^{-1} (x_1 - X_t) dt + \sigma Q_t^{-1/2} dW_t,
> $$
> where $\Sigma_{s, t} = \int_s^t Q_r dr$.
> In the limit $\sigma \to 0$, we can solve for $X$
> $$
> X_t = x_1 + \Sigma_{t, 1}\Sigma_{0, 1}^{-1}(x_1 - x_0),
> $$
> so
> $$
> \tfrac{d}{dt}X_t = Q_t \Sigma_{0, 1}^{-1}(x_1 - x_0).
> $$
> Now, if $Q_t = Q_0$ is independent of time, then
> $$
> Q_t \Sigma_{0, 1}^{-1} = Q_t \left(\int_0^1 Q_s ds\right)^{-1} = Q_0 \left(\int_0^1 I ds\right)^{-1} Q_0^{-1}  = I.
> $$
> Thus, the resulting vector field is simply $x_1 - x_0$, as in CFM.
>
> Nevertheless, if $Q_t$ does depend on time, the zero-noise bridge limit is genuinely different from that given by standard Brownian motion.
> However, such a process would be time-inhomogeneous, making a well-motivated choice unclear---in contrast to $dX_t = -cLX_t dt + \sigma dW_t$, which is physically motivated as heat equation perturbed with Brownian noise.
> Nevertheless, we could attempt a derivation in similar flavor to TFM, by choosing
> $$
> Q_t = \exp(-\kappa Lt).
> $$
> In spectral coordinates, this yields the conditional vector field
> $$
> (u_t(y)^{y_0, y_1})^i = \begin{cases}
> \frac{\kappa \lambda^i \exp(-\kappa \lambda^i t)}{1 - \exp(-\kappa \lambda^i)}(y_1^i - y_0^i) & \lambda^i > 0 \\\\
> y_1^i - y_0^i & \lambda^i = 0.
> \end{cases}
> $$
> This is a remarkably similar formula to ours, except that it lacks a topology-dependent scaling of y_0^i itself, which TFM does via $\exp(-\lambda^i \kappa)y_0^i$. If we consider a condition path going from noise $x_0$ to data $x_1$, this method essentially results in denoising lower-frequency components before higher-frequency ones.
>
> In order to further motivate our method, and compare it in a controlled setting against this newly derived method based on a topology-aware noising process, we performed a synthetic experiment on the triangulated torus.
> It is a matching task between distributions over signals on edges, where both distributions have different qualitative features: the initial one is divergence-free, exhibiting no sinks and sources, while the final one is curl-free, exhibiting no "swirls" in the sample vector fields.
> We can see that TFM outperforms CFM and TAN (which stands for topology-aware noise) in terms of Wasserstein distance.
> We can also look more closely at the spectral density of the distributions induced by each model, compared to the spectral density of the true initial and final distribution.
> Notably, we find TFM is able to convert better between the two different spectra, while CFM and TAN similarly struggle at dampening high-frequency curl-free components.
> This is clearly seen by looking at the visualised samples from each model.
>
> We deeply appreciate this fascinating topic brought up by the reviewer. We believe the synthetic experiment is a great addition to our work, as being able to compare the spectra in a controlled way makes it clearer why TFM may outperform CFM. Moreover, the research direction of topology-aware noise is very interesting---while the forms we investigated lead either directly to CFM or to a framework that does not perform as well as TFM, it does lead to valid FM frameworks and could provide a way to inject topological information in addition to the drift. Indeed, we believe this is a direction of interest to future research.

---

> ### Author Response · Authors · 2025-11-27
> **Response to Reviewer 8d47 Part 3**
>
> ## The role of Figure 2 (Weakness 4)
> We appreciate the reviewer's feedback regarding Figure 2.
>
> Our intention with the figure was to demonstrate the smoothing effect induced by the topological reference process in TFM on the conditional paths, as compared to CFM, which we found to be best conveyed visually in the image case.
> We found that this is best conveyed in the image case.
> Thus, the figure is not intended to demonstrate an advantage of one method over the other, rather only how they differ qualitatively.
> We have updated the image description specifying what the image is illustrating.
>
> We believe this improves the clarity of the paper and appreciate the reviewers attention to detail in bringing this up.
>
>
> ## Details about the hyperparameter $\kappa$ (Question 2)
> We thank the reviewer for
> Indeed, the role of the parameter kappa is important, describing the speed of the diffusion or, equivalently, the rate of exponential decay of non-zero eigenvalues.
> The higher kappa is, the higher the rate of decay and thus the larger the smoothing bias.
>
> While we deliberately did not extensively optimise the hyperparameter kappa, showing that a simple choice of $\kappa = 2.0$ already improves the performance for experiments from TSBM, we appreciate that there is considerable value in studying its effect on performance.
> To this end, we repeated the experiments from TSBM on a range of $\kappa$ values explored to showcase how the model performance depends on $\kappa$ and, therefore, if there exists an optimal choice of $\kappa$. The results are reported in Appendix F.3, which shows that the performance of TFM, and thus its advantage over CFM, can be further improved by optimizing $\kappa$, except possibly in the experiment with earthquake magnitudes, where $\kappa = 2.0$ is already close to optimal. Moreover, except for the single-cell experiment, we find that the performance vs $\kappa$ curve is approximately convex, suggesting that an optimal choice may be unique.
>
> Our choice of $\kappa$ is now made explicit in Section 5 with a reference to further study in Appendix F.3. We believe that this addition further showcases the impact of our method, for which we thank the reviewer.
>
>
> ## Conclusion
> We thank the reviewer for the encouraging assessment and for insightful questions that motivated several improvements.
> Our revisions strengthen the theoretical clarity, expand the motivation for the reference process, and provide a detailed analysis of the role of $\kappa$, which we believe greatly enhance the paper’s presentation.
> We hope these additions clarify the contributions and potential impact of our work, and we would be grateful if the reviewer considered updating their evaluation.

---

### Official Review · Reviewer_kNj3 · 2025-10-31

**Soundness:** 3
**Presentation:** 3
**Contribution:** 3
**Rating:** 8
**Confidence:** 3

**Summary:**

This paper proposes a flow matching approach to generative modelling over nodes, edges, triangles, and so on via the use of Hodge-Laplacian-based linear conditional vector fields. They theoretically build up to this through a Schrodinger bridge formulation and eigendecomposition analysis of the Hodge-Laplacian. They offer two variants of their approach – I-TFM (independent coupling) and OT-TFM (mini-batch OT coupling). Overall, they demonstrate empirical improvements over prior work: modestly for image generation and more significantly for generation on irregular topologies.

**Strengths:**

1. Good results: on images, and better on irregular grids.
2. Good theoretical build-up.
3. Two variants: I-TFM and OT-TFM.
4. Generality of Hodge-Laplacian bodes well for future extensions.

**Weaknesses:**

1. In section 2.1 in the Graphs paragraph, the subscript notation was a bit confusing for me. After reading the Simplicial complexes paragraph, I understood. But I think the flow could be improved.
2. Didn't see an ablation of different prior choices, e.g., standard Gaussian vs. heat Gaussian.
3. No experiments for $k \geq 1$.

**Questions:**

1. Can you elaborate a bit more on why we had to go through the Schrodinger bridge problem formulation to end up with the result? Why can't you remain in the deterministic setting of flow matching, or make use of the probability flow ODE?
2. Why is I sometimes better than OT and vice versa?

---

> ### Author Response · Authors · 2025-11-27
> **Response to Reviewer kNj3 Part 1**
>
> We thank the reviewer for the thoughtful review, recognising the strength of our results and theoretical developments, as well as potential of their extensions. We appreciate the apt suggestions for improving flow and adding content through ablations, and the thought-provoking questions, which we address below.
>
> ## Confusing subscript notation in the "graphs" paragraph (Weakness 1)
> We thank the reviewer for this feedback.
>
> We chose this notation in an effort to unify the notation in the "graphs" and "simplicial complexes" paragraphs. To improve the flow of ideas, we have added a note relating simplicial notation to the standard graph theory notation, and have also made our intention of unification explicit in the "graphs" paragraph.
>
> We believe this makes the presentation more approachable, for which we thank the reviewer. We welcome any further feedback.
>
> ## Ablation of different prior choices, e.g., standard Gaussian vs. heat Gaussian. (Weakness 2)
> We thank the reviewer for recommending that such an ablation be added.
>
> We have tested the performance of CFM and TFM models across the standard Gaussian and heat Gaussian initial distributions on the experiments with earthquake magnitudes and traffic flows. The results are reported in Appendix F.2 and summarized as in the table below.
> | Dataset     | Initial Dist. | I-TFM            | OT-TFM           | I-CFM            | OT-CFM           |
> |-------------|----------------|------------------|------------------|------------------|------------------|
> | **Traffic** | Normal         | $1.30_{\pm0.01}$ | $1.28_{\pm0.01}$ | $1.72_{\pm0.01}$ | $1.59_{\pm0.01}$ |
> |             | GP             | $1.27_{\pm0.01}$ | $1.27_{\pm0.00}$ | $1.47_{\pm0.01}$ | $1.45_{\pm0.01}$ |
> | **Earthquakes** | Normal     | $5.35_{\pm0.07}$ | $5.49_{\pm0.10}$ | $8.37_{\pm0.05}$ | $8.25_{\pm0.06}$ |
> |             | GP             | $4.93_{\pm0.06}$ | $5.53_{\pm0.02}$ | $7.02_{\pm0.07}$ | $7.39_{\pm0.17}$ |
>
> We believe this improves the clarity of our work and thank the reviewer for the suggestion.
>
> ## Lack of experiments for $k \ge 1$. (Weakness 3)
> We perform two experiments with $k=1$. As indicated in Table 4, these are the experiments with ocean currents and traffic flows, where signals are defined on edges of 2-simplicial complexes. We do not examine tasks with signals on simplices of dimension strictly greater than 1, as such datasets are significantly more uncommon. Additionally, face signals on a 2-simplicial complex correspond to vertex signals on the dual 2-simplicial complex; thus, a meaningfully different setting would have to consider face signals on a simplicial complex of at least dimension 3, which are even more uncommon. We hope this has addressed the reviewer's concern and welcome further discussion.
>
> ## Is SBP needed in the derivation of TFM? (Question 1)
> We thank the reviewer for this deep question.
>
> Since TFM is a framework with a deterministic vector field, solving an exact, rather than entropic, OT problem, it is possible that a derivation contained to the deterministic setting exists.
> A key challenge is that the SBP is generally not well-posed if $\sigma=0$, which we now explicitly explain in Section 3.1.
> Nevertheless, the equivalence between Monge-Kantorovich and Benamou-Brenier formulations of OT shows us that the vector field learned by TFM is the unique solution of the optimal control problem
> $$
> \min \int_0^1 \tfrac{1}{2} ||u_t(x)||^2 \, \mathbb{P}_t(\mathrm{d}x)\, \mathrm{d}t,
> \qquad \text{s.t.} \quad
> \partial_t \mathbb{P}_t + \nabla \cdot \big(\mathbb{P}_t ( H_t(\mathbf{L}_k) + \alpha_t + u_t )\big) = 0,
> \qquad \mathbb{P}_0 = \mu_0,\quad \mathbb{P}_1 = \mu_1.
> $$
> as is now explained in Section 3.1 (for CFM) and Section 4 (for TFM). Thus, one could begin a derivation of a topology-aware extension of flow matching, by positing that it should solve this problem.  However, we do not know whether such a derivation could be done without transposition to a stochastic viewpoint, and, if it could be, whether it would make the motivation clearer and the mathematics simpler than the SBP perspective. The SBP perspective offers a compelling motivation as posterior inference given a topological prior and makes the mathematics quite straight forward. Indeed, the addition of Brownian motion regularizes the problem, allowing us to use standard formulas for the bridges and the optimal transport cost.
>
> An approach based on the probability flow ODE also appears attractive, since it is provided in Topological Schrodinger Bridge Matching (Appendix E.4). However, the formula provided is given implicitly, in terms of two random variables determined by a coupled system of heat equations. This makes it computationally intractable and a derivation of TFM would require its reformulation, likely in the form given in our work.
>
> We believe the additions to our manuscript mentioned above significantly improve the presentation and motivation of our work, for which we thank the reviewer.

---

> ### Author Response · Authors · 2025-11-27
> **Response to Reviewer kNj3 Part 2**
>
> ## Performance of OT and I variants. (Question 2)
> We appreciate the reviewer's thought-provoking question.
>
> As of now, we do not have a clear answer, as the relative performance of I and OT variants appears problem-dependent.
> Indeed, since both methods learn the same final time marginals, any theoretical statements about the quality of modeling using either I or OT are challenging.
> However, it is possible that an optimization of batch size could improve the performance of the OT variants, since in the generation experiments and in the ocean current experiment we use a mini-batch approximation to the true OT coupling.
> Further investigation of this matter would certainly be of interest to future work.
>
>
> ## Conclusion
> We thank the reviewer for the thoughtful comments and for highlighting both the strengths and the future potential of our work.
> The revisions addressing notation, prior ablations, the role of SBP, and the interpretation of I vs. OT variants have, we believe, further improved the clarity and accessibility of the paper.

---

### Author Response · Authors · 2025-11-27
**Global Response**

We thank all reviewers for their careful reading and constructive feedback. We are encouraged by the positive assessments of our contributions across methodology, theory, and experiments:

(1) Strong empirical performance across diverse domains.
- kNj3: “Good results: on images, and better on irregular grids.”
- 8d47: “The method is shown competitive on various datasets… showing much potential.”
- Dei1: “experiments are conducted on several topological datasets…”
- g1EL: “the experiments are diverse.”

(2) A novel and general framework for topology-aware flow matching.
- kNj3: “Generality of the Hodge-Laplacian bodes well for future extensions.”
- 8d47: “tackles an important problem of extending FM to general geometry…”
- Dei1: “a novel generative framework for data embedded in graphs.”

(3) Clear theoretical development and valuable conceptual link between SBP and FM.
- kNj3: “Good theoretical build-up.”
- 8d47: “a beautiful analogue between the propositions and the classical results”
- Dei1: “The math is carefully derived.”
- g1EL: “unifying framework between flow matching and Schrödinger bridge matching”

At the same time, the reviews shared key areas for improvement:

(1) **Further motivation behind the topological reference process.**
Reviewer g1EL requested deeper elaboration of topology-related aspects, and reviewer 8d47 asked for clearer justification of the reference process. In response we added:
- **Motivation behind $H_t$.** Section 3.2 now clarifies that choosing $H_t(\mathbf{L}_k)$ as a polynomial ensures tractability and the ability to approximate any analytic function of $\mathbf{L}_k$ (per reviewer Dei1's question).
- **Comparison to CFM in spectral coordinates.** Addressing reviewer Dei1’s question, we discuss why CFM in spectral coordinates does not recover topology-awareness of TFM and show its lack of performance gain over standard coordinates in Appendix F.1.
- **Alternative topological reference process and synthetic experiment.** Following reviewer 8d47's suggestion (also relevant to reviewer g1EL’s request for qualitative evaluation), we derived a flow matching framework from the topology-aware reference process $d X_t = \exp(-\kappa \mathbf{L}_k t)d W_t$. We evaluated it in a synthetic experiment on a triangulated torus, comparing how TFM, CFM, and this noise-based alternative transform a divergence-free distribution into a curl-free distribution, by analysing the spectra of the predicted distributions and visually evaluating their samples. TFM yields higher fidelity and handles high-frequency components significantly better.

(2) **Additional explanation of the SBP viewpoint of FM.**
Reviewers kNj3 and Dei1 requested further justification for the SBP perspective in the derivation of TFM, as well as the elaboration on the zero-noise SBP. In response, we augmented the paper with additional explanations:
- **SBP with $\sigma=0$ is ill-posed.** Section 3.1 now gives an explicit example showing that an SBP with $\sigma=0$ is generally not well-posed, motivating the need for the limiting perspective.
- **The zero-noise limit of the SBP solution yields an optimal control path.** In Section 3.1, we note that the drift of the SBP solution solves the Benamour--Brenier OT problem. Analogously, in Section 4, we highlight that the control of the topological SBP solution solves an optimal control problem. Thus, it is the minimum-energy corrective force needed to steer from the initial to the final distribution under the heat equation dynamics.
- **Conditional control $u_t^{x_0, x_1}$ not based on the topological SBP may ignore the topology.** As per reviewer Dei1's suggestion, Section 4 now explicitly explains why the conditional control $u_t^{x_0, x_1}(X_t) = (x_1 - x_0) - (H_t(\mathbf{L}_k)X_t + \alpha_t)$ of the topological flow ODE yields a conditional path that ignores the topology.

(3) **The choice of hyperparameter $\kappa$ and its effect on performance.**
Reviewers 8d47 and g1EL raised questions about the choice and optimization of the parameter $\kappa$. In response, we made the following additions to our work:
- **Reporting of $\kappa$ in the main text.** The Experiments section now states that $\kappa$ is fixed to 0.01 for CIFAR-10 and to 2.0 for all datasets from Yang (2025).
- **Qualitative effect of $\kappa$.** We now explicitly mention in Section 3.2 that the strength of the topology-aware smoothing bias induced by the topological reference process is proportional to $\kappa$.
- **Ablation across $\kappa$ values.** To assess sensitivity $\kappa$ we ran an ablation study across a wide range of values on all Yang (2025) datasets. The results are now included in Appendix F.3. In general, we find that the performance of our method can be further improved by an appropriate choice of $\kappa$.

We believe that the revisions made in response to the reviewers' feedback have greatly improved the clarity and motivation of our work. We sincerely appreciate the reviewers' contribution.

---

### Author Response · Authors · 2025-12-03
**Summary of Contributions and Rebuttal**

We thank the AC for overseeing the paper under the exceptional circumstances of this year’s review process. We briefly summarize our contributions and key rebuttal updates.

Topological flow matching (TFM) is a topology-aware extension of conditional flow matching (CFM) for generative modelling of signals on graphs and simplicial complexes. To derive TFM, we interpret CFM as solving the zero-noise limit of a Schrödinger Bridge Problem (SBP), and augment the usual Brownian reference with a topological drift. We show that TFM offers significant improvements over CFM and topological Schrödinger bridge matching (TSBM) across real-world datasets.

During rebuttal, we strengthened the paper via improved exposition, clarified theoretical motivation, and multiple added experiments:
### Improved exposition and theoretical motivation
We clarified the meaning of “topology” in our setting by linking simplicial complexes to their geometric realizations, and gave concrete examples of how they are constructed in practice, thereby determining the Hodge Laplacian (Section 2.1). We also added an explanation of why SBP with $\sigma=0$ is ill-posed, motivating our limiting viewpoint (Section 3.1).

We clarified why naive conditional controls can cancel the topological drift, collapsing to standard CFM, whereas the SBP-based formulation yields the minimum-energy solution to a topological optimal-control problem (Section 4). We explained that choosing $H_t$ as a polynomial of the Laplacian is both tractable and expressive (Section 3.2), and we compared our drift-based design to an alternative topology-aware formulation based on a $Q$-Brownian reference process, showing why our choice is more principled and empirically effective (Appendix D).
### New experiments and ablations
- __$\kappa$-ablation__ (Appendix F.3): tuning $\kappa$ further increases gains over baselines.
- __Gaussian vs heat-Gaussian priors__ (Appendix F.2): heat-Gaussian improves performance.
- __CFM in spectral vs standard coordinates__ (Appendix F.1): no benefit from change of coordinates.
- __Synthetic torus experiment__ (Appendix D): TFM transforms spectral energy between curl-free and divergence-free distributions more faithfully than baselines.
- __Runtime comparison__: TFM is $\approx 9\times$ faster than TSBM.

## Reviewer-Specific Notes
Regrettably, the reviewers did not get the opportunity to respond before the discussion was halted. We addressed all weaknesses (W) and questions (Q) posed, as summarized below.

---
### Reviewer kNj3 (score 8)
(W1) Clarified relationship between simplicial-complex and graph notation.

(W2) Added ablation on prior choices.

(W3) Noted that __two experiments use $k = 1$__.

(Q1) Explained why SBP with $\sigma = 0$ is not well-posed and why probability-flow ODE in Yang (2025) is unsuitable for FM; discussed possibility of optimal-control derivation.

(Q2) Discussed why I- vs OT-variants may differ in practice.

---
### Reviewer 8d47 (score 4)
(W1) Added assumptions to Propositions 2,3,4 and to Section 3.1.

(W2) __Derived FM framework based on topology-aware noise and explained why it lacks the physical motivation of the Laplacian drift.__

(W3) __Added synthetic torus experiment demonstrating advantage of TFM over CFM and FM based on topology-aware noise.__

(W4) Clarified purpose of Figure 2 (illustrating the smoothing bias).

(Q2) Added $\kappa$-ablation and clarified its interpretation as the strength of topology-aware smoothing.

---
### Reviewer Dei1 (score 2)
(W1) Added explanation of topological features of simplicial complexes.

(W2) Clarified that $\mathbf L_k$ is fully determined by the simplicial complex $K$; gave examples of constructing $K$.

(W3) Added derivation showing why naive control ignores the topology.

(W4) __Explained that CFM is Euclidean and is a baseline in all experiments.__

(W5) Added empirical comparison of CFM with spectral vs standard coordinates, showing no effect on performance.

(Q1) Explained why SBP with $\sigma = 0$ is not well-posed.

(Q2) Justified polynomial drift via tractability and expressivity.

(Q3) Connected TFM on manifolds to functional models (Appendix C).

(Q4) Cited rectified flow.

---
### Reviewer g1EL (score 4)
(W1) Discussed equivalence between modeling marginal distributions of deterministic and stochastic dynamics; noted limitations of trajectory-related tasks used in prior work.

(W2) Expanded discussion of topology-related results and why the SBP viewpoint is needed to select a topology-aware control.

(Q1) (Appendix B.1) __Clarified distinction from Yang (2025): TFM provides simulation-free training, deterministic sample paths, and topology-aware initial distribution; Yang (2025) does not discuss topological FM.__

(Q2) Added $\kappa$-ablation, clarifying its fixed choice in experiments; __measured TFM to be $\approx 9\times$ faster than TSBM; showed advantage of TFM besides test metrics through spectral and visual analysis in the synthetic torus experiment.__

---

### Meta-Review · Area_Chair_fr4E · 2026-01-21

**Summary:**

Many reviewers raise question about the necessity of the theoretical derivation (Reviewer Dei1, kNj3, 8d47). This might be entangled with writing issues brought up by reviewers Dei1, reviewer 8d47, and reviewer g1EL. Reviewer g1EL concerns about novelty comparing to prior works such as Yang 2025. Reviewers Dei1 and 8d47 also concerns about experimental results, such as marginal improvement over baselines (raised by Reviewer 8d47), additional computational (reviewer 8d47), comparisons with CFM with spectral coordinates (Reviewer Dei1), and comparison with the independent-coupling baseline (Reviewer kNj3).

**Reviewer Concerns:**

The authors provided very thorough responses including improved exposition and detailed explanation in both discussion and the revision of the paper. The authors also provide several additional experiments to address the concerns about comparison with baseline and computational efficiency.

**Reviewer Scores:**

i believe reviewer kNj3 is likely to maintain his/her score. Reviewer g1EL and 8d47 might find their concerns sufficiently addressed by the rebuttal; thus I hypothesize reviewer g1EL might start to lean positive. The most negative reviewer Dei1 might also agree to raise score given that most of his/her addressed are either acknolwedge or addressed during the rebuttal.

---

### Decision · Program_Chairs · 2026-01-26

Accept (Poster)